# Targeting CDK2 overcomes melanoma resistance against BRAF and Hsp90 inhibitors

Alireza Azimi[1,†], Stefano Caramuta[1,†], Brinton Seashore-Ludlow[2,†], Johan Boström[3] (iD),
Jonathan L Robinson[4], Fredrik Edfors[5], Rainer Tuominen[1], Kristel Kemper[6], Oscar Krijgsman[6] (iD),
Daniel S Peeper[6] (iD), Jens Nielsen[4] (iD), Johan Hansson[1], Suzanne Egyhazi Brage[1], Mikael Altun[3] (iD),
Mathias Uhlen[5] (iD) & Gianluca Maddalo[5,*] (iD)

## Abstract

Novel therapies are undergoing clinical trials, for example, the Hsp90 inhibitor, XL888, in combination with BRAF inhibitors for the treatment of therapy-resistant melanomas. Unfortunately, our data show that this combination elicits a heterogeneous response in a panel of melanoma cell lines including PDX-derived models. We sought to understand the mechanisms underlying the differential responses and suggest a patient stratification strategy. Thermal proteome profiling (TPP) identified the protein targets of XL888 in a pair of sensitive and unresponsive cell lines. Unbiased proteomics and phosphoproteomics analyses identified CDK2 as a driver of resistance to both BRAF and Hsp90 inhibitors and its expression is regulated by the transcription factor MITF upon XL888 treatment. The CDK2 inhibitor, dinaciclib, attenuated resistance to both classes of inhibitors and combinations thereof. Notably, we found that MITF expression correlates with CDK2 upregulation in patients; thus, dinaciclib would warrant consideration for treatment of patients unresponsive to BRAF-MEK and/or Hsp90 inhibitors and/or harboring MITF amplification/overexpression.

**Keywords** CDK2; Hsp90 and BRAF inhibitors; melanoma; MITF; proteomics
**Subject Categories** Cancer; Genome-Scale & Integrative Biology; Post-translational Modifications, Proteolysis & Proteomics
**Mol Syst Biol. (2018) 14: e7858**

## Introduction

Malignant melanoma has the highest somatic mutational rate among cancers (Alexandrov *et al*, 2013) and its incidence is steadily increasing (https://training.seer.cancer.gov/melanoma/intro/). In the majority of the cases, it harbors BRAF (~60%) or NRAS (~20%) mutation. Other notable genomics alterations include the tumor suppressor CDNK2A deletions, KIT aberrations, MITF amplifications (10–20%), PIK3 mutations, loss of NF1 (Whittaker *et al*, 2013) and PTEN deletions (Chin *et al*, 2006).

The introduction of BRAF inhibitors (BRAFi), such as vemurafenib and dabrafenib, as standard of care for the treatment of BRAF-mutant melanoma has significantly improved the response rate among patients (Chapman *et al*, 2011). However, the benefits from BRAFi monotherapy are only temporary, as after ~6 months the majority of the patients experience tumor progression. Drug resistance results from a plethora of mechanisms, both MAPK-dependent and MAPK-independent. For example, several mutations in the MAPK pathway have been detected in BRAFi-resistant cell lines or patients' tumors, such as activating mutations in MEK1/2 (Wagle *et al*, 2014) and in NRAS (~21%; Nazarian *et al*, 2010; Shi *et al*, 2014). Resistance can proceed through switching between RAF isoforms, expression of alternative RAF splice variants (Poulikakos *et al*, 2011), alternative BRAF protein with a duplicated kinase domain (Kemper *et al*, 2016), or overexpression of CRAF (Montagut *et al*, 2008) or COT (Johannessen *et al*, 2010). MAPK-independent mechanisms have also been observed, such as upregulation of compensatory receptor tyrosine kinases, for example, PDGFRβ (Nazarian *et al*, 2010), IGF-1R (Villanueva *et al*, 2010) or EGFR (Girotti *et al*, 2013).

In 2011, the Food and Drug Administration approved the use of immunotherapy for the treatment of metastatic melanoma. The

1  Department of Oncology-Pathology, Karolinska Institutet, Karolinska University Hospital, Stockholm, Sweden
2  Chemical Biology Consortium Sweden, Science for Life Laboratory, Division of Translational Medicine and Chemical Biology, Department of Medical Biochemistry and Biophysics, Karolinska Institutet, Stockholm, Sweden
3  Science for Life Laboratory, Division of Translational Medicine and Chemical Biology, Department of Medical Biochemistry and Biophysics, Karolinska Institutet, Stockholm, Sweden
4  Department of Biology and Biological Engineering, Chalmers University of Technology, Göteborg, Sweden
5  Science for Life Laboratory, School of Biotechnology, KTH Royal Institute of Technology, Stockholm, Sweden
6  Division of Molecular Oncology & Immunology, The Netherlands Cancer Institute, Amsterdam, The Netherlands
   *Corresponding author. Tel: +46 708 93 56 54; E-mail: gianluca.maddalo@scilifelab.se
   †These authors contributed equally to this work

response rate of patients treated with immunotherapy is generally lower than targeted therapy (10–40% versus ~60%), although the clinical benefit of immunotherapy is often more stable (~2 years) and patients with tumors lacking BRAF mutations can also benefit from such treatment (Flaherty *et al*, 2010; Topalian *et al*, 2012; Johnson & Puzanov, 2015). Unfortunately, one of the main side effects of immunotherapy is the development of immune-related adverse events, which in some cases can be fatal. Thus, there is an urgent need for new drug strategies that would prevent the development of melanoma resistance to targeted therapies and would be suitable for patients that are inherently unresponsive to targeted- or immunotherapies currently in clinical use.

One such strategy to attenuate the development of resistance is the use of combinations of drugs (Al-Lazikani *et al*, 2012). A number of combination therapies are undergoing clinical trials and have been reported to overcome melanoma primary/acquired resistance to drug treatments, such as Hsp90 inhibitors (Hsp90i; e.g., XL888; Phadke *et al*, 2015). In this particular case, the underlying rationale is to inhibit the chaperone activity of Hsp90, which assists in the folding of several oncoproteins, such as AKT, CDK4, COT, ERBB2/3, FYN, or CRAF (Taipale *et al*, 2012), whose upregulation underlies the onset of resistance (Paraiso *et al*, 2012). Hsp90i have been reported to overcome acquired resistance to BRAF and MEK inhibitors in melanoma cell lines (Smyth *et al*, 2014). The combined therapies vemurafenib-XL888 (Paraiso *et al*, 2012; NCT01657591) and the recent triple combination BRAFi-MEKi-XL888 (NCT02721459) are currently being tested in clinical trials in patients with melanoma harboring BRAF mutations. To date, a set of protein biomarkers that would enable monitoring of patient response and distinguish between responders and non-responders to these combined therapies is not available. The only attempt to generate such a shortlist of potential biomarkers was performed by Rebecca *et al* (2014) on BRAF- or NRAS-mutated responsive cell lines/patient specimens.

Importantly, when we assayed cell viability on a panel of melanoma cell lines that included PDX-derived disease models, a subset was unresponsive to Hsp90i, pointing to an urgent need for patient stratification strategies. To make matters worse, the spectrum of molecular (off-) targets of Hsp90i has not been thoroughly investigated. The off-targets might cause a paradoxical activation of mechanisms of resistance to the drug therapy as was shown previously for the BRAFi PLX4032 (Poulikakos *et al*, 2010).

In this study, we aimed at providing a systems-level understanding of the differential response to BRAFi and Hsp90i classes of inhibitors and their combinations in sensitive and non-responsive cell lines. We employed a thermal proteome profiling (TPP) approach to investigate the protein targets of the Hsp90i XL888 in sensitive and resistant cell lines and tease apart their eventual differences in terms of drug targets so as to provide insight into the mechanisms of resistance to XL888. In parallel, we employed orthogonal unbiased proteomics and phosphoproteomics approaches, which offer a systems-level understanding of the cell signaling pathways that contribute to the inherent unresponsiveness to Hsp90i and BRAFi classes. The results provided by these complementary approaches enabled us to design drug strategies to overcome melanoma resistance to both BRAFi and Hsp90i monotherapies and their combination. Our *in vitro* findings would warrant consideration for more in-depth *in vivo* studies.

## Results

### Heterogeneous response to BRAFi and Hsp90i in a panel of melanoma cell lines

Given the current clinical trials testing BRAFi and Hsp90i, we sought to identify a drug therapy that would overcome both BRAFi and Hsp90i inherent resistance simultaneously. In order to understand factors influencing drug response to the single treatments, we first assessed the cell viability with an MTS assay upon treatment with dabrafenib in a panel of BRAF-mutant melanoma cell lines that included patient-derived xenografts (PDX) collected before treatment with vemurafenib (M026.X1.CL) and after the onset of resistance due to an acquired NRAS mutation (M026R.X1.CL; Possik *et al*, 2014; Kemper *et al*, 2016). Six out of nine cell lines were resistant to dabrafenib: three were inherently unresponsive (SK-Mel 24, SK-Mel 28, and ESTDAB037), whilst three were unresponsive due to acquired resistance developed either *in vitro* (A375 DR1 and MNT-1 DR100) or *in vivo* (M026R.X1.CL; Fig 1A).

Next, in the same panel we assessed the cell viability following XL888 treatment and investigated whether any resistant cell line might be inherently unresponsive to this Hsp90i currently used in clinical trials to treat melanoma (NCT01657591). Interestingly, XL888 drastically reduced the viability in the previously unresponsive SK-Mel 24 cells, but not in SK-Mel 28, or in cells with acquired resistance to BRAFi (M026R.X1.CL, MNT-1 DR100, A375 DR1; Fig 1B). Notably, the parental cell lines A375 and MNT-1 (which are sensitive to BRAFi) did not respond to XL888 treatment.

Since SK-Mel 24 and SK-Mel 28 were both unresponsive to BRAFi, but showed differential apoptotic response to BRAFi-XL888 combined therapy or XL888 monotherapy (Fig 1C), we decided to further investigate the underlying differences between these two cell lines.

---

**Figure 1.  Different cell responses upon treatment with BRAF and Hsp90 inhibitors.**

A   Cell viability measured on a panel of melanoma cells upon 72-h treatment with dabrafenib (BRAFi) (±SD is plotted; *n* = 3).

B   Cell viability measured on a panel of melanoma cells upon 72-h treatment with XL888 (Hsp90i) (±SD is plotted; *n* = 3).

C   Analysis of apoptotic cells in SK-Mel 24 and SK-Mel 28 by annexin V after 72-h treatment with BRAFi (1 μM dabrafenib) or Hsp90i (200 nM XL888) alone and BRAFi (1 μM dabrafenib) and Hsp90i (200 nM XL888) combined treatment (±SD is plotted; *n* = 3).

D   Cell viability for SK-Mel 28 (left panel) and SK-Mel 24 (right panel) was measured after 72-h treatment with other Hsp90 inhibitors (aside from XL888), such as AUY022, BIIB021, novobiocin, and 17-DMAG. (Discrepancies in the concentration–response profiles for SK-Mel 28 to XL888 between (B and D) can be attributed to the viability surrogate that is measured. In the case of (B), the MTS assay, quantifying metabolic activity, is used, whereas (D) is based on readout of ATP using CellTiter-Glo. In both assays, incomplete killing occurs at the concentrations of drugs used) (±SD is plotted; *n* = 3).

E   ZIP model to evaluate the combined effect BRAFi and Hsp90i in SK-Mel 28 (±SD is plotted; *n* = 3).

F   ZIP model to evaluate the combined effect BRAFi and Hsp90i in SK-Mel 24 (±SD is plotted; *n* = 3).

▶

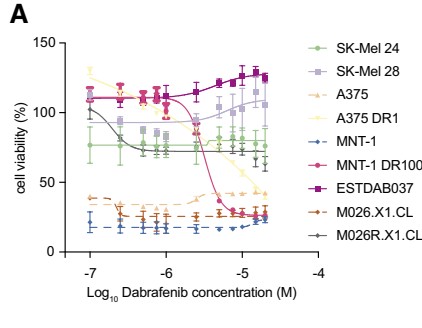

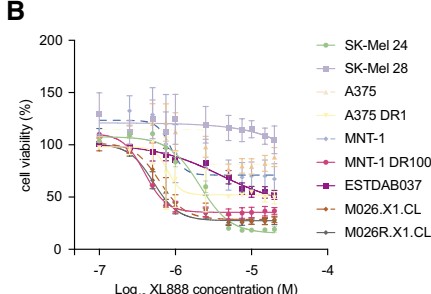

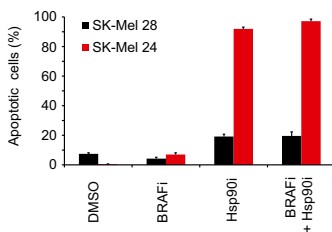

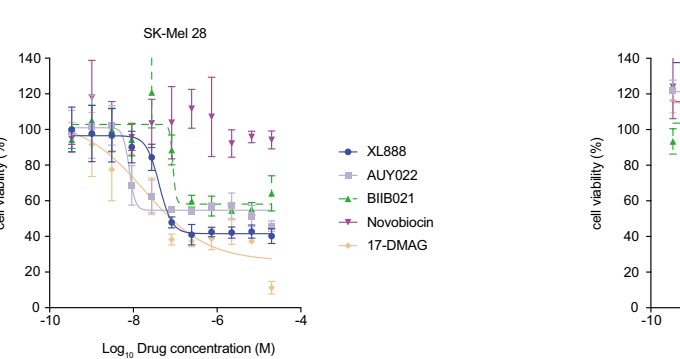

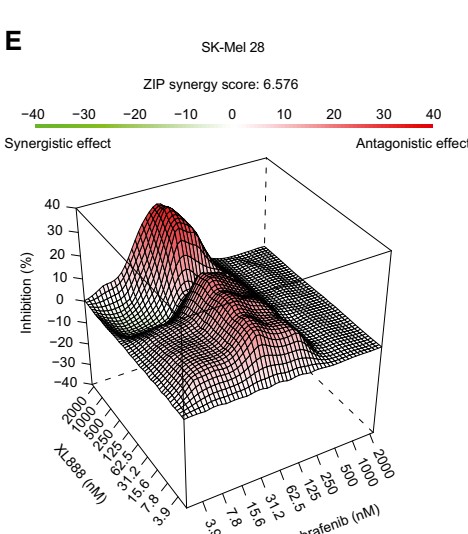

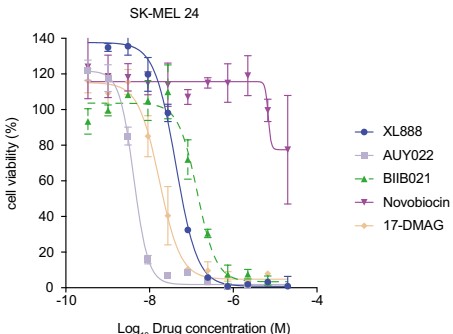

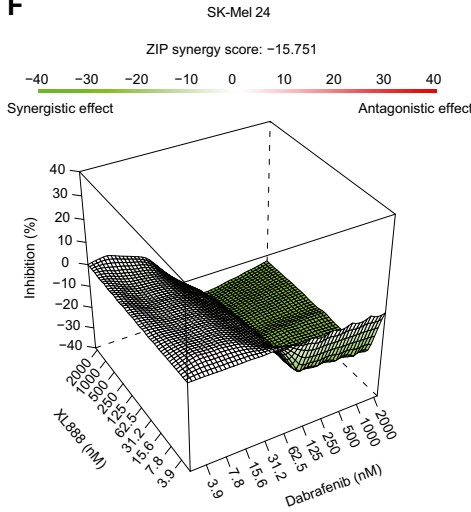

**Figure 1.**

In order to determine whether the cell lines were unresponsive to XL888 specifically or to Hsp90i in general, SK-Mel 28 and SK-Mel 24 were treated with a panel of Hsp90i (including AUY022, BIIB021, novobiocin, and 17-DMAG) and ATP levels were measured as a surrogate for cell viability. Our results indicate that treatment with the Hsp90i panel, with the exception of novobiocin, yielded complete loss of viability in SK-Mel 24 (Fig 1D, right panel), while in SK-Mel 28 the viability was only partially reduced (Fig 1D, left panel).

To understand the results from the combined treatment (1 μM dabrafenib plus 200 nM XL888 from Fig 1C), we performed full isobologram treatment of dabrafenib and XL888 in the differentially responsive cell lines, SK-Mel 28 and SK-Mel 24. A zero interaction potency (ZIP) model was used to interpret the results (Yadav *et al*, 2015), revealing that in SK-Mel 28 an antagonistic response is observed for some of the concentrations of XL888 combined with dabrafenib (Fig 1E; e.g., 1 μM dabrafenib plus 200 nM XL888), indicating that the cell viability is affected to a lesser extent as compared to monotherapy for this cell line. In SK-Mel 24 cells, however, a synergistic effect is detected in the combined treatment (Fig 1F). Overall, these melanoma cell lines, while being inherently resistant to BRAFi, responded heterogeneously to both Hsp90i monotherapy and the combined treatment BRAFi-Hsp90i. The results highlight the urgent need for patient stratification strategies for monotherapies with BRAFi and Hsp90i and their combination, especially when antagonism between the two drugs is observed in some cases.

## TPP and phospho-TPP reveal different protein targets of the Hsp90i XL888 in sensitive and unresponsive cells

To elucidate the mechanisms underlying a differential response of melanoma cells to BRAFi, Hsp90i, and their combined treatment, we employed a multifaceted proteomics approach. We first investigated the binding partners of XL888 in SK-Mel 24 and SK-Mel 28 cells using thermal proteome profiling (TPP; Martinez Molina *et al*, 2013; Savitski *et al*, 2014). We performed TPP in lysate and in intact cells with and without XL888 treatment for the two cell lines (Fig EV1A–C). The TPP experiments and analyses are summarized in Table 1 and Fig EV1D (Dataset EV1).

Here, we found that Hsp90 and GLUD2 are the main targets shared between SK-Mel 24 and 28 from the two experimental layouts (drug/DMSO). Overall, the two cell lines show poor overlap in terms of drug–protein engagement. Regarding the (few) shared entries in the intact cell layout (drug/DMSO), we observed stabilization of the chaperone CDC37, a known Hsp90 interactor (Taipale *et al*, 2014; Dataset EV1, Fig EV1E).

Moreover, for the first time, we performed the analysis of the thermal stability of the phosphoproteome in intact cells, providing a snapshot of the cell signaling response due to the drug treatment perturbation. Here, we observed higher thermal stability of phosphorylated CRAF (pS301) in SK-Mel 28 (drug/DMSO; Dataset EV1).

Importantly, aside from comparing treated versus control samples in different layouts, it is also possible to investigate the eventual inherent differences of the proteome and phosphoproteome thermal stability of sensitive and unresponsive cell lines to gain insight into the signaling pathways that underlie the resistance of SK-Mel 28 to Hsp90i (Fig EV1F). Overall, we built an interaction map based on the statistically significant proteins/phosphoproteins (Figs EV2 and EV3). Considering the "druggability" of the kinome, we focused our

attention on protein kinases with differential thermal stability between SK-Mel 28 versus SK-Mel 24 cells using the cell extract layout. The comparison revealed differences among regulators of cell cycle progression, for example, CDK2, CDK4, CDK6, and PAK2, highlighting these proteins as potential targets to overcome the inherent resistance of SK-Mel 28 (Fig EV4A). The higher thermal stability of CDK6 was observed also in the intact cells layout (Dataset EV1).

Notably, when comparing the phosphoproteome thermal stability of the two cell lines, we observed higher stability in SK-Mel 28 of the protein regulator of cell proliferation pPAK4 (S474; which regulates the expression of the transcription factor MITF, responsible for melanocyte differentiation; Yun *et al*, 2015), and the transcription factor pSTAT1 (S727; Fig EV4A). The higher thermal stability of the (active) kinase pPAK4 (S474) was also confirmed by Western blot (Fig EV4B). This suggests that the active state of these two phosphorylated proteins might be more stable in the resistant cells due to additional post-translational modifications.

Overall, the different comparisons of the (phospho)proteome thermal stability of resistant versus sensitive cells in different settings using TTP identified six potential targets, whose inhibition could sensitize the resistant cells to XL888 (henceforth referred as Hsp90i): CDK2, CDK4, CDK6, CRAF, PAK2, PAK4, and STAT1.

## Proteomics and phosphoproteomics analyses reveal that primary resistance to BRAFi is accompanied by MAPK pathway activation

As a complementary approach to strengthen the initial hypotheses gained from TPP, we employed an unbiased genomewide proteomics and phosphoproteomics platform to address a number of relevant biological questions in SK-Mel 24 and SK-Mel 28. This provided insight into changes of the membrane proteome that may have been missed using TPP. Cell lines that were sensitive (SK-Mel 24) or unresponsive (SK-Mel 28) to XL888 treatments were grown under conditions as detailed in Fig 2A for 48 h, lysed, and the lysates digested by Lys-C/trypsin. Protein quantification was performed using a label-free approach. The same workflow was

**Table 1.  Results of the TPP investigations in different settings.**

| TPP | | No. of proteins |
|---|---|---|
| Lysate | SK-Mel 24 Hsp90i/DMSO | 50 |
| | SK-Mel 28 Hsp90i/DMSO | 48 |
| | SK-Mel 28 DMSO/SK-Mel 24 + DMSO | 61 |
| | SK-Mel 28 Hsp90i/SK-Mel 24 Hsp90i | 71 |
| In-Cell | SK-Mel 24 Hsp90i/DMSO | 42 |
| | SK-Mel 28 Hsp90i/DMSO | 39 |
| | SK-Mel 28 DMSO/SK-Mel 24 DMSO | 67 |
| | SK-Mel 28 Hsp90i/SK-Mel 24 Hsp90i | 64 |

| Phospho-TPP | | No of phosphopeptides |
|---|---|---|
| In-Cell | SK-Mel 24 Hsp90i/DMSO | 27 |
| | SK-Mel 28 Hsp90i/DMSO | 80 |
| | SK-Mel 28 DMSO/SK-Mel 24 DMSO | 49 |
| | SK-Mel 28 Hsp90i/SK-Mel 24 Hsp90i | 35 |

used for the analysis of the phosphoproteome with the addition of a TiO$_2$ phosphopeptide enrichment step. Overall, we identified ~7,000 proteins and ~15,500 phosphosites with a localization probability > 0.75 (Dataset EV2 and EV3). The quality of our data was assessed by a principal component analysis (PCA) that showed a clear partition between the two cell lines and the different treatments employed in this study (Fig EV4C).

In both cell lines, we observed upregulation of Hsp70 upon treatment with XL888 or BRAFi-XL888 (Dataset EV1). Both SK-Mel 28 and SK-Mel 24 are inherently unresponsive to BRAFi (Fig 1A), and in both cases, the MAPK pathway was activated. In particular, upon BRAFi treatment SK-Mel 24 cells show upregulation of pERK1 (pY204) and pERK2 (pY187) at phosphoproteome level (Dataset EV3). Moreover, our data show upregulation at proteome level of DCLK, a protein involved in tumorigenesis (Hayakawa *et al*, 2017), and JAK1, a protein involved in melanoma resistance against BRAFi (Kim *et al*, 2015; Dataset EV2). Similarly, in SK-Mel 28 we observed upregulation of pERK2 (pT185, pY187) and pCRAF (pS296) at phosphoproteome level (Dataset EV3), as well as upregulation of CDK2 at proteome level (Dataset EV2). CRAF activation/upregulation has been previously reported to be involved in mechanisms of resistance to BRAFi (Montagut *et al*, 2008) and was also observed in the TPP experiments (see previous section). The activation of the MAPK pathway upon BRAFi treatment in both cell lines was confirmed by Western blot (Fig EV4D).

### Inhibition of PAK1, PAK4, and CDK2 overcome the resistance to XL888

To gain insight into the differential Hsp90i response, we compared unresponsive (SK-Mel 28) versus sensitive (SK-Mel 24) cells. However, as the two cell lines are not isogenic, we compared each cell line (upon treatment with Hsp90i) to its untreated control (DMSO) and subtracted the upregulated entries of the sensitive cells from those of the resistant cells (Fig 2B, upper panel). This subtractive analysis generated a protein list that was analyzed using GOrilla GO (Eden *et al*, 2009), which revealed an enrichment of proteins involved in detoxification, as well as development of pigmentation (Dataset EV4). Among the upregulated proteins, our analysis

identified nine kinases, three of which were "druggable": CDK2, PAK1, and PAK4 (Fig 2B, lower panel). Note that CDK2 and PAK proteins were also identified in our TPP analyses (see previous section), further supporting their involvement in the resistance to XL888.

The same subtractive approach was applied to analyze the phosphoproteome in the same setting (Hsp90i/DMSO) and the phosphopeptides that were uniquely upregulated in SK-Mel 28 were analyzed by kinase-substrate enrichment analysis (KEA; www.maayanlab.net/KEA2; Casado *et al*, 2013) to predict the most upregulated kinase activities. This bioinformatics analysis predicted CDK2 and GSK3β (*P*-value < 0.05 and intersected genes > 7) to be the main kinases involved in the phosphorylation events (Fig 2C).

Overall, the results generated by our proteomics and phosphoproteomics platforms combined with TPP analyses provided a number of potential targets whose inhibition might sensitize SK-Mel 28 to XL888: CDK2; CDK4; CDK6; CRAF; PAK1, PAK2; PAK4; STAT1; and GSK3β. This shortlist of potential targets was enriched in CDK and PAK proteins; CDKs are involved in cell cycle progression and are often dysregulated in cancer (Malumbres & Barbacid, 2009). Similarly, PAK proteins are generally upregulated in cancer and are involved in cell survival and angiogenesis, controlling several processes that are implicated in cancer initiation (Radu *et al*, 2014). We assayed the cell viability of our model cell line SK-Mel 28, resistant to XL888 and BRAFi, using specific inhibitors alone and in combination with Hsp90i against each candidate, and among these, dinaciclib (CDKi), FRAX597 (PAK1/2/3i), and PF-3758309 (PAK4i; together with Hsp90i) reduced the cell viability below 50% (Fig 2D).

### Targeting CDK2 sensitizes unresponsive cells to both BRAFi and Hsp90i classes individually and in combination

Considering that our main goal was to design a drug therapy with high potential to be used in clinics to overcome both BRAFi and Hsp90i resistances and their combination, we investigated the resistance to BRAFi to decide which of these three drugs (CDK2i, PAK1i, PAK4i) to analyze further with other cell lines. We exploited the same subtractive rationale shown previously in Fig 2B; we compared each cell line upon treatment with BRAFi to its relative

**Figure 2.  Proteomics and phosphoproteomics findings.**

A   Scheme of the different settings (BRAFi = 1 μM dabrafenib; Hsp90i = 200 nM XL888; BRAFi+Hsp90i = 1 μM dabrafenib + 200 nM XL888) employed in this study after 48 h treatment.

B   Venn diagram of the upregulated protein entries in sensitive (SK-Mel 24) and resistant (SK-Mel 28) cells upon treatment with 200 nM XL888 (Hsp90i/DMSO) after 48 h (*n* = 3). The 240 unique entries for SK-Mel 28 are highlighted in red, among which nine are protein kinases (upper panel). Volcano plot generated by the comparison between Hsp90i/DMSO in SK-Mel 28 after 48 h (*n* = 3; lower panel).

C   Venn diagram of the upregulated phosphopeptides in sensitive (SK-Mel 24) and resistant (SK-Mel 28) cells upon treatment with 200 nM Hsp90i (Hsp90i/DMSO) at 48 h (*n* = 3). The 534 unique phosphopeptides for SK-Mel 28 were analyzed by KEA. This bioinformatics analysis predicted CDK2 and GSK3β as upstream active kinases.

D   Effects on the cell viability after 72 h of the inhibitors (and their combinations) that target the potential entries reported in the text for SK-Mel 28 (BRAFi = 1 mM dabrafenib; Hsp90i = 200 nM XL888; CDK2i = 200 nM dinaciclib; GSK3βi = 2 μM CHIR-99021 HCl; PAK1/2i = 2 μM FRAX597; PAK4i = 2 μM PF-3758309; STAT1i = 2 μM Fludarabine; CDK4/6i = 2 μM palbociclib). The red arrows highlight the settings were the cell viability falls below 50% upon drug treatment (±SD is plotted; *n* = 3).

E   The same rationale used in (A) was exploited for the proteomics analysis of the effects of BRAFi (1 μM dabrafenib) treatment (upper panel) after 48 h (*n* = 3). Volcano plot generated by the comparison between BRAFi/DMSO in SK-Mel 28 after 48 h (*n* = 3; lower panel).

F   The same rationale used in (A) was exploited for the proteomics analysis of BRAFi-Hsp90i combined therapy after 48 h (*n* = 3).

G   Overlap of the downregulated (upper panel) and upregulated (lower panel) kinases at proteomics level unique for SK-Mel 28 in different settings at 48 h (*n* = 3). In red, the only shared "druggable" upregulated kinase CDK2 is highlighted.

H   Western blot analysis confirms the upregulation of CDK2 in different settings (upper panel). Band intensities were normalized against the mean of β-actins, and lane 1 was used as reference (lower panel).

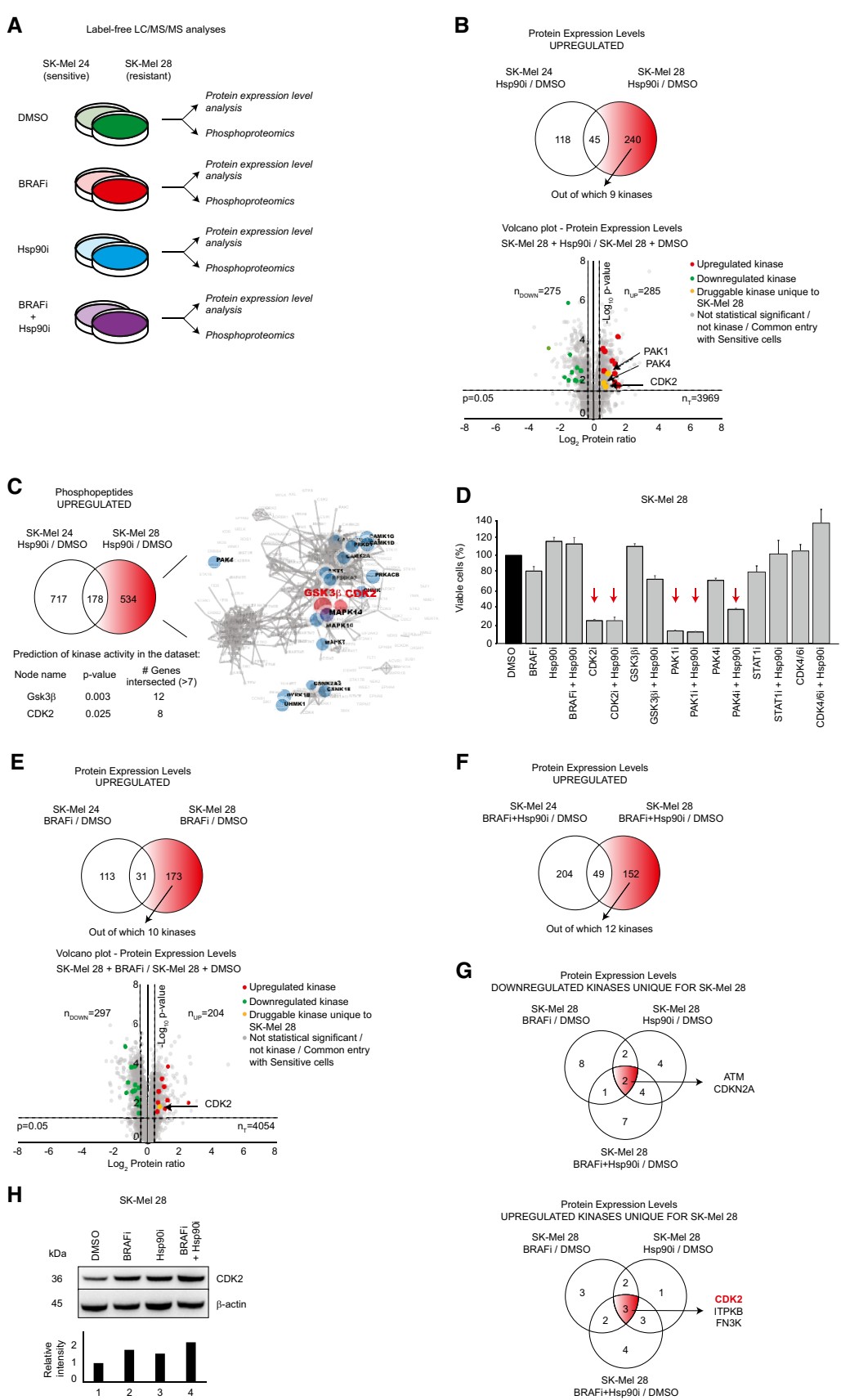

**Figure 2.**

control (DMSO), and we subtracted the upregulated entries of the sensitive cells from the resistant ones (Fig 2E). Considering the "druggability" of the kinome, we focused our attention on the upregulated kinases and a shortlist of ten candidates was generated (Fig 2E). Again, CDK2 was one of the main druggable targets.

We applied the same subtractive strategy for the analysis of SK-Mel 28 undergoing the combined treatment BRAFi-Hsp90i/DMSO and a shortlist of twelve kinases was generated (Fig 2F).

Finally, we compared these shortlists of kinases uniquely down-regulated in SK-Mel 28 in the three different settings (BRAFi/DMSO, Hsp90i/DMSO, and BRAFi-Hsp90i/DMSO), and the overlapping analyses retrieved two proteins involved in regulation of cell cycle progression, for example, ATM and CDKN2A (Fig 2G, upper panel). Similar analyses on the upregulated entries revealed three shared upregulated kinases, for example, CDK2, ITPKB, and FN3K, among which only CDK2 was "druggable" (Fig 2G, lower panel). Notably, CDK2 was the only kinase that showed an opposite trend in expression levels between sensitive and resistant cells upon BRAFi-Hsp90i treatment (Fig EV4E). Our proteomics data showing the upregulation of CDK2 in SK-Mel 28 in different settings was confirmed by Western blot analysis (Fig 2H).

Based on these results, we proceeded with further investigations using dinaciclib. Importantly, this CDK2i is already used in clinical trials against leukemia (phase III; NCT01580228) and melanoma (phase II; NCT00937937); hence, its toxicity in patients has been already assessed favorably (Ghia *et al*, 2017) and it has high potential to enter clinics.

Considering that dinaciclib is a pan-CDK inhibitor targeting CDK1, CDK2, CDK5, and CDK9 (Parry *et al*, 2010), we assayed SK-Mel 28 cell viability using a panel of specific CDK inhibitors (Table 2), providing insight into which kinase would play a major role in the viability of Hsp90i unresponsive cells. The data clearly show that only the inhibition of CDK2 has an effect on cell viability, which was potentiated by the simultaneous treatment with Hsp90i (Fig 3A). Based on these results, we generated a shRNA doxycycline-inducible knockdown against CDK2 (Fig 3B). The conditional knockdown of CDK2 sensitizes the cells to Hsp90i (Fig 3C, left panel) and BRAFi (Fig 3C, right panel) treatments (Fig EV4F), clearly showing that CDK2 is a key player underlying melanoma resistance to both Hsp90i and BRAFi.

### Cell lines resistant to Hsp90i and BRAFi are sensitive to dinaciclib

We assayed the cell viability against dinaciclib (henceforth referred as CDK2i) in a panel of 11 BRAF-mutated cell lines, including two

**Table 2.    IC50 of the CDKs inhibitors employed in this study.**

|  | IC50 | | | |
|---|---|---|---|---|
|  | CDK1 | CDK2 | CDK5 | CDK9 |
| Ro 3306[a] | 20 nM | | | |
| K03861[b] | | 15.4 nM | | |
| Roscovitine[c] | | 0.7 μM | 0.16 μM | |
| LDC000067[d] | 5.5 μM | 2.4 μM | | 44 nM |

[a]Vassilev LT, *et al Proc Natl Acad Sci USA* 2006, 103(28), 10660–10665.
[b]Alexander LT, *et al ACS Chem Biol* 2015, 10(9), 2116–2125.
[c]Meijer L, *et al Eur J Biochem* 1997, 243(1-2), 527–536.
[d]Albert TK, *et al Br J Pharmacol* 2014, 171(1), 55–68.

PDX-derived cell pairs, obtained before BRAFi treatment, M026.X1.CL and M029.X1.CL, and after treatment, upon tumor relapse, M026R.X1.CL and M029R.X1.CL, respectively (Fig 4A; Possik *et al*, 2014; Kemper *et al*, 2016). Dinaciclib was effective against all the employed cell lines, reducing the cell viability below 50% (Fig 4A). We also evaluated the effect on cell viability when adding dinaciclib to the combination therapy BRAFi-MEKi (used as first-line therapy in clinics against melanoma), as well as other single and combined treatments. Our data show that BRAFi-MEKi-CDK2i triple treatment (together with CDK2i-Hsp90i combined therapy) is the most effective among the different strategies that were tested and does not generate any antagonistic effect. Notably, the BRAFi-MEKi-Hsp90i currently used in clinical trials (NCT02721459) was unable to reduce the cell viability below the threshold of 50% in SK-Mel 28 and ESTDAB 37, unlike BRAFi-MEKi-CDK2i or CDK2i-Hsp90i.

Similarly, CDK2i potentiates the effect of MEKi in two NRAS-mutated cell lines, SK-Mel 2 and ESTDAB 102 (Fig 4B). The most potent apoptotic effect is observed in ESTDAB 102 when dinaciclib is combined with XL888 (Fig 4C).

### MITF is the master regulator of Hsp90i resistance through CDK2 upregulation

To gain insight into the transcription factors that govern the response to Hsp90i, we conducted a bioinformatics prediction analysis of the 240 significantly upregulated proteomics entries that were unique to SK-Mel 28, which is unresponsive to Hsp90i treatment (*P*-value < 0.05 and fold change ≥ 1.5; Fig 5A). These were analyzed using ChEA (http://amp.pharm.mssm.edu/Enrichr), a manually curated database from which the over-representation of transcription factors in a dataset is predicted (Lachmann *et al*, 2010). This analysis identified MITF, a transcription factor responsible for melanocyte differentiation (Cheli *et al*, 2010), as the only statistically significant entry (adjusted *P*-value = 1.73 e$^{-7}$; Dataset EV5). Accordingly, our data showed upregulation of a number of proteins that control the transcription of MITF such as pPAK4 (pS474), β-catenin 1 (CTNNB1; Yun *et al*, 2015) and ZEB2 (Denecker *et al*, 2014). Furthermore, we observed upregulation of a number of known downstream MITF targets, such as the marker for cell differentiation and pigmentation DCT (Guyonneau *et al*, 2004) and CDK2 (Hoek *et al*, 2008; Fig 5A). In addition, we observed upregulation of pMITF in our phosphoproteomics data. Western blot analysis confirmed upregulation of MITF and its transcriptional targets DCT and CDK2 as well as downregulation of pERK only upon Hsp90i treatment as compared to the control (DMSO; Fig 5B). The upregulation of MITF and its transcriptional target DCT are involved in melanin synthesis, an event that is clearly visible in the cell pellets (Fig 5B, bottom panel). Based on these results, we generated a doxycycline-inducible shRNA knockdown against MITF. Importantly, aside from MITF, we observed remarkably reduced expression of its transcriptional target CDK2 (Fig 5C). The knockdown of MITF sensitized SK-Mel 28 to Hsp90i, but not to BRAFi treatment, confirming that MITF is essential only for the resistance to Hsp90i treatment (Fig 5D, left panel), but not against BRAFi (Fig 5D, right panel). We assessed the upregulation of MITF and its targets DCT and CDK2 in two additional cell lines, A375 and the PDX-derived M029R.X1.CL. Our Western blot analyses confirmed clearly the same results observed for SK-Mel 28 (Fig 5E). Cell lines

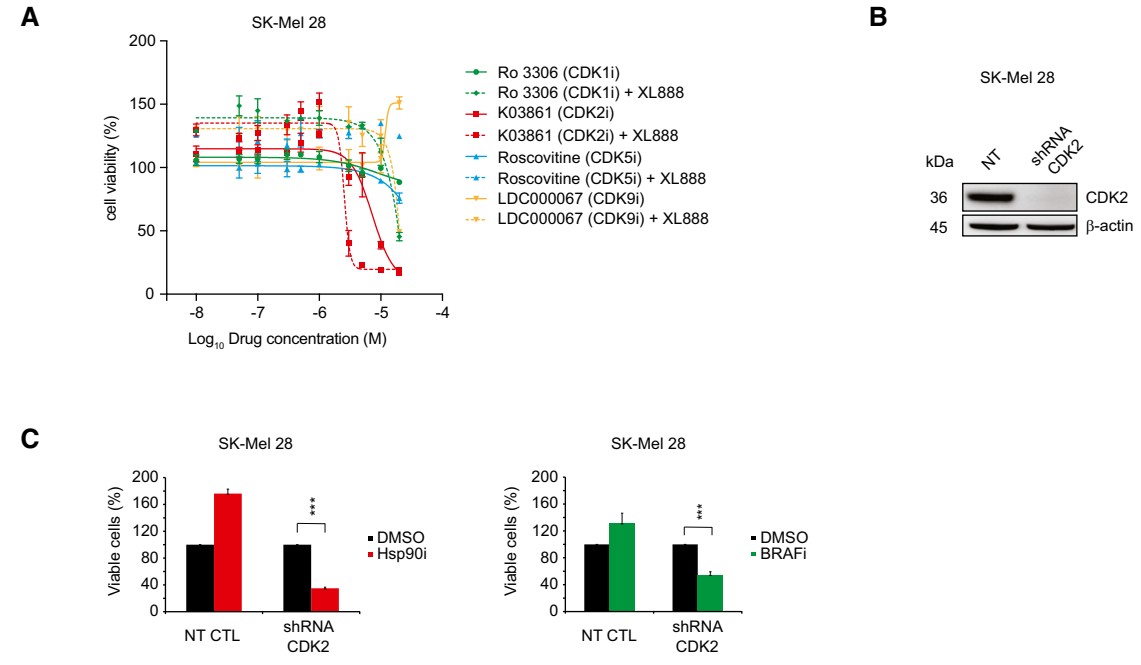

**Figure 3. Validation of CDK2 as driver of melanoma resistance against Hsp90i and BRAFi.**

A   Cell sensitivity to CDK1 (Ro 3306), CDK2 (K03861), CDK5 (Roscovitine), and CDK9 (LDC000067) inhibitors ± 200 nM XL888 (Hsp90i) at 72 h in SK-Mel 28 was analyzed by MTS assay (±SD is plotted; $n = 3$).

B   The induction by doxycycline of the conditional shRNA knocks down CDK2 expression levels at 72 h in SK-Mel 28.

C   Cell viability assay of the conditional CDK2 knockdown cell upon Hsp90i (200 nM XL888; left panel) and BRAFi (1 µM dabrafenib; right panel) at 72 h in SK-Mel 28 ($n = 3$; 72–96 h ± doxycycline and 72 h inhibitors; $t$-test $P$-value < 0.001) (±SD is plotted; $n = 3$).

that are sensitive to Hsp90i show instead a downregulation of the expression levels of CDK2 (Fig EV4G).

Overall, our data reveal that in our employed model cell line SK-Mel 28, the inherent resistance to BRAFi is accompanied by activation of the MAPK pathway, which promotes cell cycle progression and CDK2 activation. On the other hand, upon Hsp90i treatment, cell survival is driven by MITF, which sustains CDK2 upregulation, whilst the MAPK pathway is inactive (Fig 5B and F). Overall, in both cases, there is a reliance on CDK2.

**CDK2 and MITF expression levels correlate in melanoma cell lines and patients**

To further validate our findings, we probed the correlation of MITF and CDK2 in the publicly available Cancer Cell Line Encyclopedia (CCLE) containing 935 cell lines. Among the different cancer types, melanoma-derived cell lines exhibited the highest and most significant correlation (Pearson $r = 0.8$, $P_{adj} = 2.4 \ e^{-12}$) between MITF and CDK2 mRNA expression levels (Fig 6A, Dataset EV6). Of the melanoma cell lines employed in this study, the highest co-expression values for MITF and CDK2 were observed in SK-Mel 28.

To corroborate this finding in patient samples, we performed the same analysis using The Cancer Genome Atlas (TCGA) transcriptomics data. We observed a similar positive correlation between MITF and CDK2 in melanoma patients (Fig 6B), where melanoma exhibited the greatest and most significant correlation (Pearson $r = 0.6$, $P_{adj} = 3.3 \ e^{-53}$) among the available cancers (Dataset EV6).

Furthermore, the correlation between high expression of MITF and CDK2 was also confirmed by IHC analyses of melanoma patients (Fig 6C) from the publicly available data from Protein Atlas (www.proteinatlas.com; Dataset EV7).

Overall, our orthogonal analyses point to CDK2 as a candidate whose inhibition overcomes inherent resistance to BRAFi and Hsp90i (driven by MITF) and their combination in melanoma.

## Discussion

Considering that the list of driver oncogenes is enriched in protein kinases (Fleuren et al, 2016) and that there is a growing body of evidence showing that, in some cases, cancer does not harbor any genetic mutation (Mack et al, 2014; Parker et al, 2014; Versteeg, 2014), it is imperative to integrate genomics analyses with independent and orthogonal proteomics and phosphoproteomics investigations. Surprisingly, the number of phosphoproteomics studies is still underrepresented compared to genomics studies.

Here, we performed a multi-layer study on the Hsp90i XL888, which is used together with BRAFi in clinical trials to treat melanoma. We identified the protein targets of Hsp90i in intact cells and lysate layouts in differentially responsive cell lines using TTP. Notably, by comparing the thermal stability of the proteome and phosphoproteome of sensitive and unresponsive cells, we gained insight into the potential targets (CDK2 and PAKs) that would explain the different response to XL888 treatment.

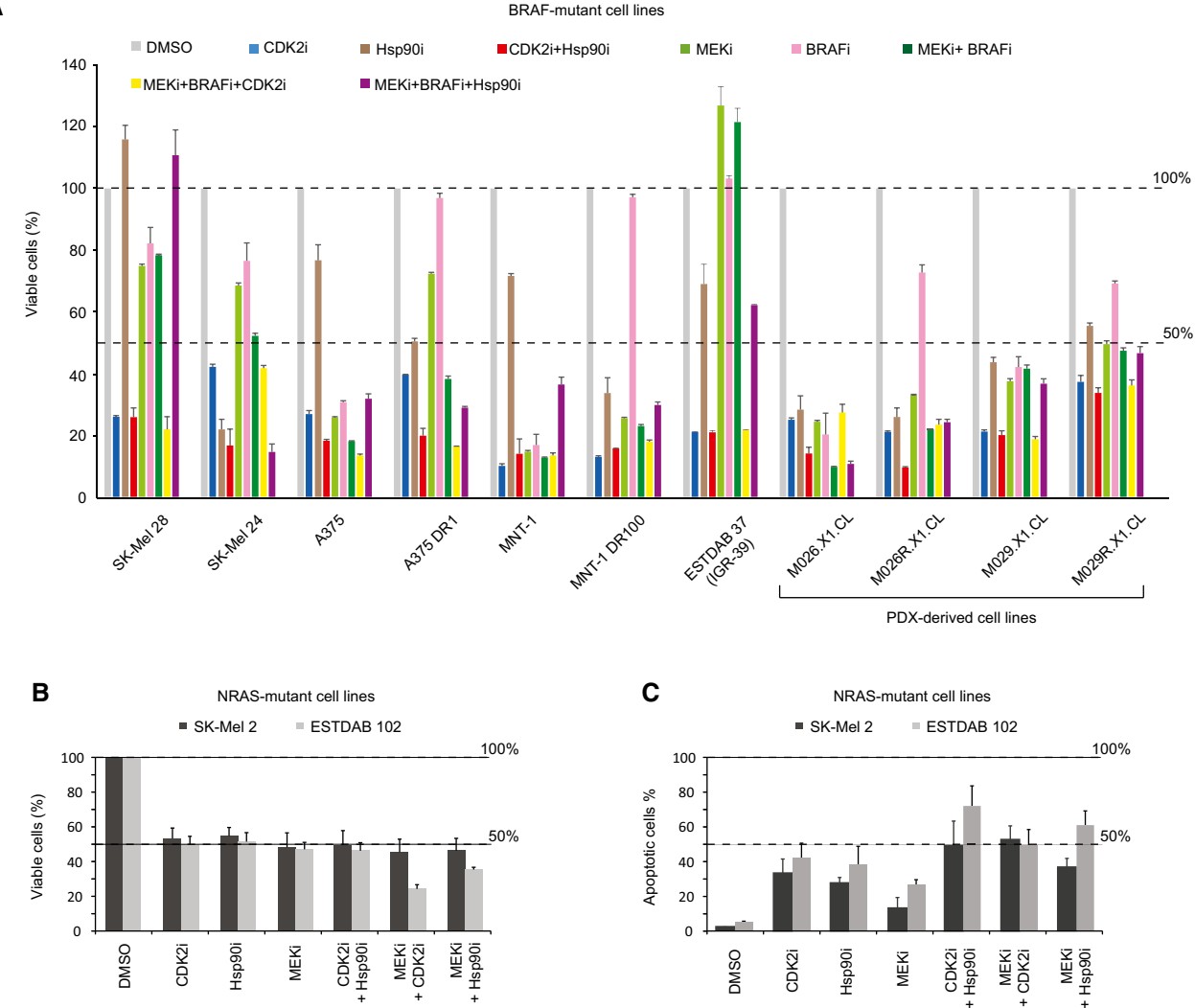

**Figure 4. Dinaciclib overcomes drug resistance in multiple cell lines.**

A Cell viability was measured on a panel of 11 BRAF-mutated cell lines in different settings at 72 h (CDK2i = 200 nM dinaciclib; Hsp90i = 200 nM XL888; CDK2i+Hsp90i = 200 nM dinaciclib + 200 nM XL888; MEKi = 100 mM trametinib; BRAFi = 1 µM dabrafenib; MEKi+BRAFi = 100 mM trametinib + 1 µM dabrafenib; MEKi+BRAFi+CDK2i = 100 mM trametinib + 1 µM dabrafenib + 200 nM dinaciclib; MEKi+BRAFi+Hsp90i = 100 mM trametinib + 1 µM dabrafenib + 200 nM XL888) ($\pm$SD is plotted; $n$ = 3).

B Cell viability was measured in two NRAS-mutant cell lines upon different drug treatments at 48 h ($\pm$SD is plotted; $n$ = 3).

C Apoptotic status was measured in two NRAS-mutant cell lines upon different drug treatments at 48 h ($\pm$SD is plotted; $n$ = 3).

Considering that TPP is biased against membrane proteome, we pursued complementary unbiased genomewide proteomics and phosphoproteomics approaches to gain a systems-level understanding of the molecular mechanisms underlying the resistance.

To date, there is no strategy that would be able to predict (i) which patients would benefit from BRAFi-Hsp90i/BRAFi-MEKi-Hsp90i and (ii) monitor their response. The only attempt to address these issues was performed by Rebecca *et al* (2014), where the authors set up a targeted proteomics analysis to follow up ~80 proteins, mainly Hsp90 clients, to monitor patient response. However, their study presented some limitations as it was performed only on responsive cell lines (no resistant cell lines were employed in their workflow); hence, it is not evident from their work which

biomarker can be used with high(er) confidence to distinguish between responsive and unresponsive cell lines/tumors. In this regard, in our study we observed that the Hsp90 client AKT1 is downregulated in both sensitive and unresponsive cells upon Hsp90i monotherapy and BRAFi-Hsp90i combined therapy (Fig EV4H); thus, it is not necessarily a valid marker for distinguishing which patients will respond. In contrast, CDK2 is the only kinase that in our data could distinguish between responsive and unresponsive cell lines, showing different trends in terms of expression levels (Fig EV4E). Therefore, the valuable shortlist suggested by Rebecca *et al* to monitor the therapy response would need to be further refined including in the analysis additional settings (e.g., BRAFi-Hsp90i) and resistant cell lines/tumors. This

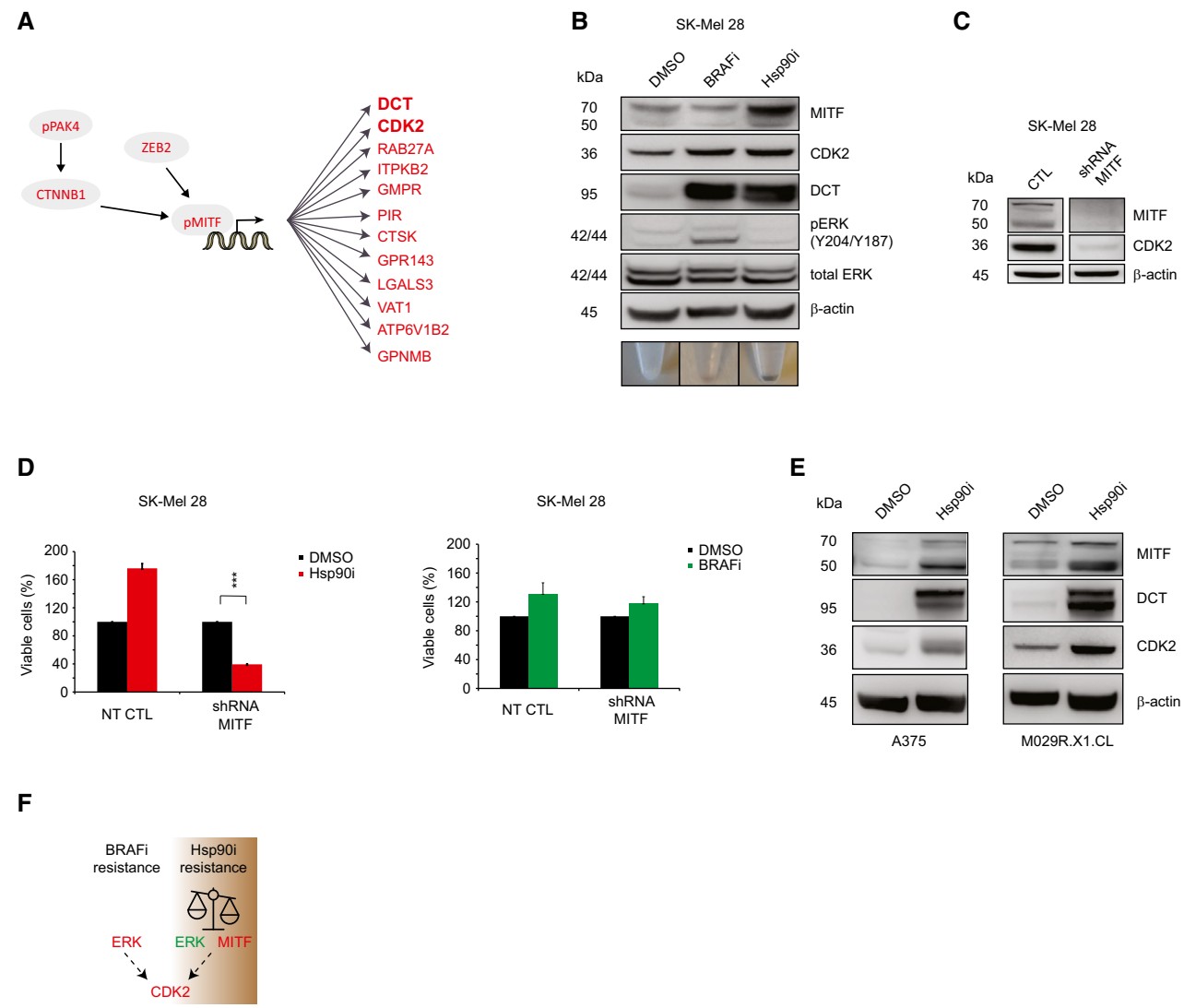

**Figure 5. MITF upregulation leads to Hsp90i resistance.**

A   Overexpressed entries (in circles) that positively regulate the transcription of MITF and its downstream transcriptional targets (in red; $P < 0.05$; fold change $\geq 1.5$) observed in our proteomics dataset ($n = 3$).

B   Western blot analyses confirmed our bioinformatics prediction analyses by ChEA. Note the inverse correlation between pERK and MITF expressions in SK-Mel 28. The color of the cell pellets of SK-Mel 28 upon treatments at 48 h with DMSO, BRAFi, and Hsp90i is shown in the lower panel.

C   The knockdown of MITF causes downregulation of CDK2 expression levels in SK-Mel 28.

D   Cell viability assay of the SK-Mel 28 MITF conditional knockdown upon Hsp90i (200 nM XL888; left panel) and BRAFi (1 μM dabrafenib; right panel) treatments at 72 h ($n = 3$; 72–96 h ± doxycycline and 72 h inhibitors; t-test ***$P$-value $< 0.001$) (±SD is plotted; $n = 3$).

E   Western blot analyses show the upregulation of MITF and its transcriptional targets CDK2 and the melanotic marker DCT upon 200 nM XL888 treatment at 72 h in A375 and M029.R.X1.CL.

F   Model of cancer plasticity that leads to resistance to both BRAFi and Hsp90i treatments by switching between different signaling pathways.

refinement will certainly benefit from the *in vivo* analyses of patient-derived material generated by the ongoing clinical trial studies (NCT01657591 and NCT02721459).

We show that the resistance to Hsp90i can be overcome by targeting different kinases (PAK1, PAK4, and CDK2) in our model system; however, in-depth analyses reveal that CDK2 is the only shared upregulated druggable kinase that governs resistance to both the BRAF and Hsp90 classes of inhibitors and the combination thereof.

We investigated the mechanisms that govern the CDK2 expression and in agreement with previous studies (Du *et al*, 2004), we showed that CDK2 expression is controlled by MITF, a transcriptional factor responsible for melanocyte differentiation (Wellbrock & Arozarena, 2015), which has been reported to be controlled by pPAK4 (pS474), β-catenin 1 (CTNNB1; Yun *et al*, 2015), and ZEB2 (Denecker *et al*, 2014).

Importantly, our study sets a remarkable example of cancer plasticity that underlies melanoma primary resistance to drug therapies: upon BRAFi treatment, the resistance is accompanied by MAPK

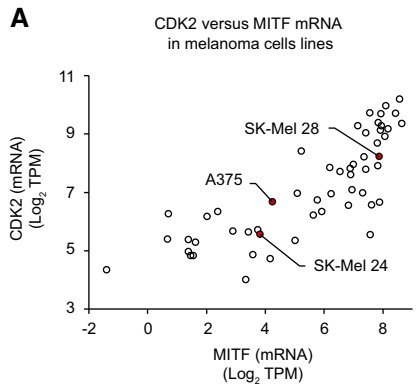

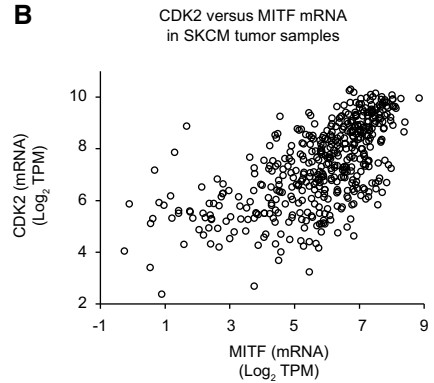

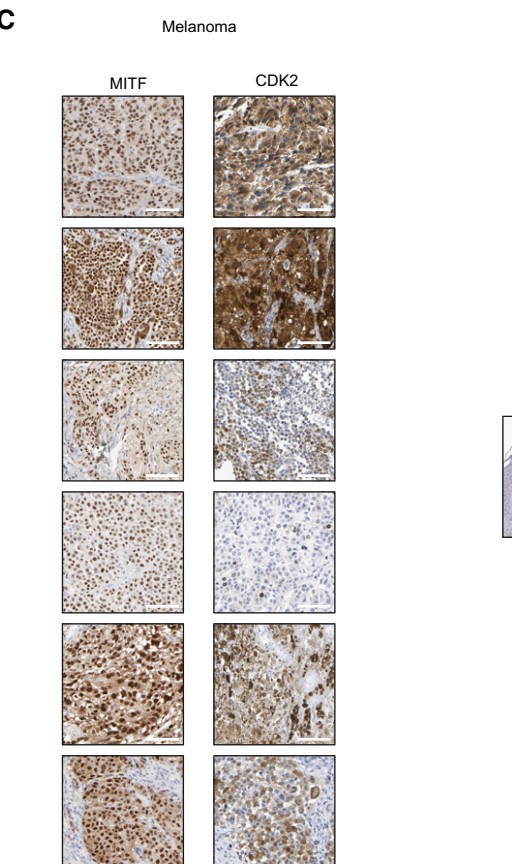

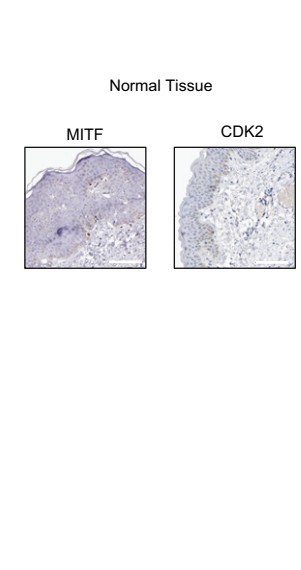

**Figure 6.  Analysis of CDK2 and MITF expression in CCLE and TGCA databases.**

A   Plot of CDK2 versus MITF mRNA abundance (log₂ TPM) among all melanoma-derived cell lines in the CCLE. Cell lines included in this study have been labeled.
B   Plot of CDK2 versus MITF mRNA expression in skin cutaneous melanoma (SKCM) patient samples from TCGA.
C   IHC images of matched patient material show strong staining of MITF and CDK2 (co)expression in melanoma tissues (left panels). Normal tissues show low or no (co)expression in skin tissue (right panels). The data were kindly provided by the Protein Atlas Project publicly available (scale bar is 100 μm) (www.proteinatlas.org).

pathway activation, which triggers cell cycle progression and CDK2 activation (Lents *et al*, 2002). For the first time, it is shown that upon Hsp90i treatment melanoma upregulates MITF expression to sustain CDK2 upregulation to survive, while the MAPK pathway is inactive. The role of MITF in melanoma is controversial as low MITF/AXL

ratio has been related to poor predictive response to targeted therapy in melanoma (Muller *et al*, 2014), while others have reported the involvement of MITF in unresponsiveness to MAPKi and melanoma progression (Wellbrock & Arozarena, 2015; Smith *et al*, 2016). Here, we show that MITF upregulation drives resistance to Hsp90i, but not

to BRAFi, by governing the expression of its transcriptional target CDK2. We show that this kinase plays a role of paramount importance in the resistance to both BRAFi and Hsp90i classes of drugs and their combination (Fig 6C). Our data are in line with Du et al (2004), identifying CDK2 as a drug target for melanomas.

Considering that MITF is amplified in ~20% of melanomas (Garraway et al, 2005), no drug is available in clinics that would directly target this transcription factor, and its expression correlates with CDK2 in melanoma patients (Fig 6B and C), our work would suggest a rationale for stratifying patients with MITF amplification/overexpression and treating them using a dinaciclib-based therapy, rather than a Hsp90i treatment. This CDK2i is already used in clinical trials (NCT01657591 and NCT02721459), it has been already shown to be tolerated by patients (Ghia et al, 2017); hence, it has high potential to be approved for clinical use.

Importantly, considering that the BRAFi-MEKi combination is already used in clinics and that XL888, as well as dinaciclib, is used in clinical trials, it seems logical to combine them as they do not generate any antagonistic effect in our data. Our in vitro results reveal that the triple treatment, CDK2i-BRAFi-MEKi, as well as the double-treatment CDK2i-Hsp90i, is effective in all employed cell lines, unlike BRAFi-Hsp90i/BRAFi-MEKi-Hsp90i used in clinical trials. Our data thus indicate that these therapies warrant consideration for further in vivo studies.

# Materials and Methods

## Chemicals and reagents

The drugs employed in this study were as follows: dabrafenib (ApexBio, B1407-50); XL888 (ApexBio, A4388-25); fludarabine (Selleckchem, S1491); CHIR-99021 HCl (CT99021) (Selleckchem, S2924); palbociclib (ApexBio, A8316); LDC000067 (ApexBio, B4754-10); Ro 3306 (ApexBio, A8885-10); roscovitine (ApexBio, A1723-10); K03861 (Selleckchem, S8100); CHIR-99021 (Selleckchem, S2924); dinaciclib (Selleckchem, S2768); FRAX597 (Selleckchem, S7271); PF-3758309 (Selleckchem, S7094); AUY922 (Selleckchem, S1069); BIIB021 (Selleckchem, S1175); novobiocin (Selleckchem, S2492); 17-DMAG (Selleckchem, S1142). All the drugs were dissolved in DMSO (Sigma D2650).

## Cell lines and culture conditions

SK-Mel 24, SK-Mel 28, A375, A375DR1 (dabrafenib resistant; 1 μM), MNT-1 (kindly provided by Dr. Pier Giorgio Natali, Istituto Regina Elena, Rome, Italy) and MNT-1-DR100 (dabrafenib resistant, 100 nM) cells were grown in Gibco Medium Essential medium (MEM; ThermoFisher Scientific) supplemented with 10% FBS (15% for SK-Mel 24), 1% non-essential amino acids, 1% sodium pyruvate, 1% penicillin and streptomycin. ESTDAB37 and ESTDAB102 [received from The European Searchable Tumour Line Database (ESTDAB)], SKMEL2, M026.X1.CL, M026R.X1.CL, M029.X1.CL, and M029R.X1.CL (post-relapse, resistant to BRAF inhibitor treatment; Possik et al, 2014; Kemper et al, 2016) were grown in Gibco RPMI 1640 (ThermoFisher Scientific) supplemented with 10% FBS, 1% non-essential amino acids, 1% sodium pyruvate, 1% penicillin and streptomycin.

## Sample preparation for TPP

### Intact cells

SK-Mel 24 and SK-Mel 28 cells were incubated in the presence of DMSO (control) or drug (100 μM) for 2 h. Cells were harvested, resuspended in 1.1 ml of HBBS (Gibco) supplemented with 20 mM $MgCl_2$, 1 mM sodium orthovanadate, 1 tablet of Complete mini EDTA-free mixture (Roche Applied Science), and one tablet of Phos-STOP phosphatase inhibitor mixture per 10 ml of lysis buffer (Roche Applied Science). The cell suspension per cell line (SK-Mel 24 and SK-Mel 28) and condition (+/− drug) was divided into ten aliquots of 100 μl and transferred into 0.2-ml PCR tubes. TPP was performed as previously described (Franken et al, 2015). Briefly, each tube was heated individually at the different temperatures for 3 min in a thermal cycler (Applied Biosystems (Foster City, CA)/Life Technologies) followed by cooling for 3 min at room temperature. Cell were lysed by freeze and thaw cycles, and the lysates were centrifuged at $20,000 \times g$ for 30 min at 4°C to separate the soluble fractions from precipitates. The supernatants were transferred to new 0.2-ml microtubes.

### Lysate

SK-Mel 24 and SK-Mel 28 were lysed in lysis buffer (HBBS (Gibco) supplemented with 20 mM $MgCl_2$, 1 mM sodium orthovanadate, 1 tablet of Complete mini EDTA-free mixture (Roche Applied Science), and one tablet of PhosSTOP phosphatase inhibitor mixture per 10 ml of lysis buffer (Roche Applied Science)) by freeze and thaw, and the lysates were centrifuged at $20,000 \times g$ for 30 min at 4°C to separate the soluble fractions from precipitates. Each cell line's supernatant was incubated with either DMSO or 100 μM drug at room temperature for 30 min. The cell suspension per cell line (SK-Mel 24 and SK-Mel 28) and condition (+/− drug) was divided into ten aliquots of 100 μl and transferred into 0.2-ml PCR tubes. TPP was performed as previously described (Franken et al, 2015).

Each supernatant for cell line (SK-Mel 24 and SK-Mel 28); condition (+/− drug); and experiment (intact cells/lysate) was reduced by 2 mM DTT at room temperature for 1 h; alkylated by 4 mM chloroacetamide for 30 min at room temperature at the dark. A first enzymatic digestion was performed using Lys-C (1:75 w/w) at 37°C overnight; a second enzymatic digestion was performed using trypsin (1:75 w/w) at 37°C overnight. One hundred μg of each sample was labeled by TMT10plex according to the manufacturer's instructions. Samples of each set (10 different temperatures) were mixed 1:1 (v/v) and cleaned by Strata™-X-C 33 μm Polymeric Strong Cation (Phenomenex). The whole procedure was executed in two biological replicates.

## Sample preparation for protein expression levels analysis

SK-Mel 24 and SK-Mel 28 were grown in four different conditions: in the presence of DMSO (control); 1 μM dabrafenib; 200 nM XL888; and 1 μM dabrafenib plus XL888200 nM each for 48 h. Cell pellets were harvested and resuspended in lysis buffer (8 M urea, 100 mM triethylammonium bicarbonate pH 8.5, 1 mM sodium orthovanadate, 1 tablet of Complete mini EDTA-free mixture (Roche Applied Science), and one tablet of PhosSTOP phosphatase inhibitor mixture per 10 ml of lysis buffer (Roche Applied Science)). Cells were then lysed by 10 rapid passages through a

23-gauge hypodermic syringe needle and by sonication on ice. After centrifugation (20,000 × *g* 30 min at 4°C), the protein concentration was determined by Bradford assay (Pierce). Proteins were reduced with 2 mM DTT at room temperature for 1 h, alkylated with 4 mM chloroacetamide at room temperature for 30 min in the dark. A first enzymatic digestion step was performed using Lys-C at 37°C for 4 h (enzyme/substrate ratio 1:50). Moreover, the sample was digested overnight at 37°C with trypsin (enzyme/substrate ratio 1:50). Peptides were desalted by reverse phase using Waters Sep-Pak 1cc (50 mg) cartridges (WAT054960; Waters, Milford, MA). The resin was rinsed with ACN and then equilibrated with 0.6% acetic acid. The samples were loaded and washed with 0.6% acetic acid and eluted with 80% ACN/0.6% acetic acid.

### Proteomics and phosphoproteomics analyses

Label-free quantification was performed by analyzing the raw data by MaxQuant (version 1.5.3.30; Cox & Mann, 2008). Andromeda (Cox *et al*, 2011) was used to search the MS/MS data against the UniProt *Homo sapiens* database (containing canonical and isoforms_42144 entries downloaded on March 21, 2016) complemented with a list of common contaminants and concatenated with the reversed version of all sequences. See Appendix for further details.

To perform a pairwise comparison and filter for those proteins/phosphopeptides that have a consistent abundance level over three biological replicates, we applied a two-sample *t*-test using Perseus 1.5.3.2 (Cox & Mann, 2008). Only those proteins that had a *P*-value < 0.05 and an arbitrary cutoff ratio ≥ 1.5 or ≤ −1.5 fold changes were considered. Only phosphopeptides with a location probability ≥ 0.75 were considered for statistical analyses.

### Proliferation assay (MTS)

CellTiter 96 AQueous One Solution Cell Proliferation Assay (MTS) was purchased from Promega (Cat. no. G3582, Promega, Madison, WI, USA).

### Flow cytometry-based immunostaining and apoptosis/necrosis analysis

To evaluate the presence of apoptosis/necrosis, we used annexin V-Fluos (cat. no. 11828681001 Roche) and propidium iodide and analyzed by NovoCyte flow cytometer (ACEA biosciences, Inc. San Diego, CA). Between 4 and $10 \times 10^4$ cells per well were cultured and treated with XL888 or dinaciclib for 48–72 h. Then, the cells were collected and rinsed in PBS, pelleted, and resuspended in incubation buffer (10 mmol/l HEPES/NaOH, pH 7.4, 140 mmol/l NaCl, 5 mmol/l $CaCl_2$) containing 1% annexin V and 1% propidium iodide (PI) for 10 min.

### Plasmid generation and creation of inducible stable cell lines

Inducible shRNA constructs were created by ligating annealed shRNA-coding oligonucleotides (see Dataset EV8) into an AgeI/EcoRI double-digested inducible shRNA vector as described previously (Eshtad *et al*, 2016). The constructs were validated using

sequencing over the shRNA insertion area. The plasmids, together with a non-targeting shRNA plasmid (Eshtad *et al*, 2016), were packaged into lentiviral particles using a third-generation lentiviral production system described previously (Dull *et al*, 1998), using $CaCl_2$-mediated transfection of HEK293T cells, and the produced lentiviral particles were used to infect SK-Mel 28 cells together with 0.4 μg/ml hexadimethrine bromide, which were then selected for successful integration with 1 μg/ml puromycin over 5 days.

### Data and software availability

- The TPP data have been deposited to the ProteomeXchange Consortium (http://proteomecentral.proteomexchange.org) via the PRIDE partner repository (Vizcaino *et al*, 2013) with the dataset identifier PXD005508.
- The phospho-TPP data have been deposited to the ProteomeXchange Consortium (http://proteomecentral.proteomexchange.org) via the PRIDE partner repository (Vizcaino *et al*, 2013) with the dataset identifier PXD005547.
- The data were analyzed using an in-house R-package using similar criteria to those previously described (Savitski *et al*, 2014). The raw data were analyzed using a similar workflow to one previously described (Franken *et al*, 2015) except that each detected peptide was fit and analyzed individually instead of at a protein level. See Appendix for further information. The script is available as "Code EV1".
- The mass spectrometry proteomics and phosphoproteomics data have been deposited to the ProteomeXchange Consortium (http://proteomecentral.proteomexchange.org) via the PRIDE partner repository (Vizcaino *et al*, 2013) with the dataset identifier PXD005518.

**Expanded View** for this article is available online.

### Acknowledgements

We acknowledge Prof. Sonia Lain and Prof. David Lane and their groups for technical support and for granting us permission to access their Orbitrap Fusion. The cell line pairs M026.X1.CL and M026R.X1.CL, and M029.X1.CL and M029.R.X1.CL have been used according to the Material Transfer Agreement (MTA) for academic institutions V01 13 and V08 16, respectively. GM has been awarded grants from: O. E. och Edla Johanssons foundation (5310-7132); Swedish Cancer Society (Radiumhemmets; 154202); and Lars Hiertas Minne. The authors acknowledge the entire staff of the Protein Atlas Project for the IHC images. We acknowledge Prof. Janne Lehtiö and Rozbeh Jafari for support with TPP.

### Author contributions

AA performed cell viability experiments, FACS analyses, validation of the MS data by Western blot analysis, and cell viability experiments using dox-inducible cell lines, designed the experiments, wrote the manuscript, and approved the final draft; SC performed drug assay experiments on multiple cell lines, designed the experiments, analyzed data, wrote the manuscript, and approved the manuscript; BS-L performed cell viability assay, isobolograms, analysis of phospho-TPP data, designed the experiments, wrote the manuscript, and approved the final draft; JB generated the dox-inducible cell lines, analyzed the data, and approved the manuscript; JLR performed analysis of CCLE and

TCGA data, contributed to the manuscript preparation, and approved the final draft; FE supplied the IHC images, contributed to the manuscript preparation, and approved the final draft; RT performed cell viability experiments, contributed to the manuscript preparation, and approved the final draft; KK and OK obtained the PDX-derived cell lines, contributed to the manuscript preparation, and approved the final draft; DSP, JN, JH, SEB, MA, MU contributed to the manuscript preparation and approved the final draft; GM performed the TPP, phospho-TPP, proteomics, and phosphoproteomics experiments, performed the data analyses, designed the study, analyzed the data, wrote the manuscript, and approved the final draft.

## Conflict of interest

The authors declare that they have no conflict of interest.

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
