## [Review Process File · Molecular Systems Biology]

Targeting CDK2 overcomes melanoma resistance against BRAF and Hsp90 inhibitors

Alireza Azimi, Stefano Caramuta, Brinton Seashore-Ludlow, Johan Boström, Jonathan L. Robinson, Fredrik Edfors, Rainer Tuominen, Kristel Kemper, Oscar Krijgsman, Daniel S. Peeper, Jens Nielsen, Johan Hansson, Suzanne Egyhazi Brage, Mikael Altun, Mathias Uhlen and Gianluca Maddalo

Review timeline:

Submission date:	2 August 2017
Editorial Decision:	18 September 2017
Revision received:	5 December 2017
Editorial Decision:	9 January 2018
Revision received:	15 January 2018
Accepted:	1 February 2018

Editor: Thomas Lemberger

Transaction Report:

1st Editorial Decision

18 September 2017

Thank you again for submitting your work to Molecular Systems Biology. We have now heard back from the two referees who agreed to evaluate your manuscript. As you will see from the reports below, the referees find the topic of your study of potential interest. They raise, however, several important points, which should be convincingly addressed in a major revision of the work.

Without repeating all the issues raised by the reviewers, the major points refer to the following:

- Clarification of the way phosphosite were counted and reported
- Generalization to multiple resistant cell lines (reviewer #2)
- Deemphasize the importance of the TPP profiling or clarify how it was exploited.

On a more editorial level, we would kindly ask you to include a formal Data and software availability section after the Materials & Methods:

#Data and software availability

The datasets and computer code produced in this study are available in the following databases:

- [data type]: [full name of the resource] [accession number/identifier] ([doi or URL or identifiers.org/DATABASE:ACCESSION])
- RNA-Seq data: Gene Expression Omnibus GSE46843
[<https://www.ncbi.nlm.nih.gov/geo/query/acc.cgi?acc=GSE46843>]
- Chip-Seq data: Gene Expression Omnibus GSE46748

[<https://www.ncbi.nlm.nih.gov/geo/query/acc.cgi?acc=GSE46748>]
- Protein interaction AP-MS data: PRIDE PXD000208
[<http://www.ebi.ac.uk/pride/archive/projects/PXD000208>]
- Imaging dataset: Image Data Resource doi:10.17867/10000101
[<http://doi.org/10.17867/10000101>]
- Modeling computer scripts: GitHub
[<https://github.com/SysBioChalmers/GECKO/releases/tag/v1.0>]

REVIEWER REPORTS

Reviewer #1:

MSB-17-7858

Targeting CDK2 overcomes melanoma resistance against BRAF and Hsp90 inhibitors
Azimi et al

Summary

The manuscript of Azimi et al tackles a very important question in the field of melanoma therapy: how to overcome intrinsic and acquired resistance to BRAF inhibitors (BRAFi) through judicious selection of other molecularly-targeted therapies that can be combined with BRAFi. This is particularly important given ongoing clinical trials that combine BRAFi with inhibitors of Hsp90 (Hsp90i). The authors therefore conduct a series of proteomic and phospho-proteomic experiments and integrate the data to identify CDK2 as a driver of resistance to BRAFi+Hsp90i. They then proceed to validate that targeting CDK2 with the inhibitor dinaciclib attenuates resistance to BRAFi and Hsp90i. Furthermore, the authors demonstrate the role of MITF in driving CDK2 upregulation, suggesting that MITF might serve as a biomarker for stratification of patients for CDK2 inhibitor therapy.

Overall, this is a very interesting paper dealing with a crucially important topic (intrinsic and acquired resistance to BRAFi and BRAFi combination therapies). At this time, however, there remain several points which the authors should address before the manuscript is suitable for publication in MSB.

Major points

1. The thermal proteome profiling (TPP) experiments are interesting, but they do not add significantly to the primary findings of Azimi et al regarding CDK2. These TPP experiments suggest CDK6, CDK4, PAK2, PAK4, and STAT1 as potential targets to sensitive SK-Mel-28 cells to HSPi (line 213 and Fig. 2F). Notably, CDK2 is absent from this list. In addition, the validation experiments in Fig 4C shows that STAT1i and CDK4/6i are not effective in sensitizing SK-Mel-28 to HSP90i (PAK4i did successfully sensitize SK-Mel-28 to HSP90i, but it was not nearly as effective as CDK2i or PAK1i, which had been found using alternative proteomic methods). Thus, TPP did not identify the primary focus of this article (CDK2) and was not extremely successful at predicting which targets would sensitive to HSP90i. Inclusion of the TPP data as a main figure thus distracts from the authors' primary finding that integrative proteomics identifies CDK2 as a driver of melanoma resistance to BRAFi and HSP90i (Fig. 4). This would be a much simpler and more effective manuscript if the TPP data were moved to the supplement and de-emphasized in the main text.
2. In the thermal protein profiling experiments (Fig. 2f), is there a statistical value that can be used to quantify the difference between the SK-Mel-28 and SK-Mel-24 curves? Some curves look clearly different (pSTAT1 S727) but others look so very slightly different (PAK2) that the authors need to provide some statistical justification for highlighting these proteins.
3. Line 225: the authors report ~24,000 phosphosites of which ~15,500 have localization probability > 0.75. This threshold of 0.75 localization probability is standard for the field of phospho-proteomics and phosphosites with lower probability are typically ignored or not even reported. The problem in Azimi et al is that from reading the main text and the figure legend, it is unclear if the authors are including the phosphosites with low localization probability in their subsequent analyses. In the supplement, however, the authors have buried the information that "Only phosphopeptides

with a location probability {greater than or equal to} 0.75 were considered for statistical analyses." In the interests of accuracy and simplicity, the authors should remove the mention of ~24,000 phosphosites and amend line 225 to read "we identified ~15,500 phosphosites with localization probability greater than 0.75." There is no reason to even mention the ~8,500 low confidence phosphosites.

4. Figure 3 and Figure 4: Figure 3 does not add much to the manuscript because it contains only i) a technical workflow; ii) PCA to show "the quality of [the] data" (line 227); and one Western blot confirming re-activation of MAPK pathways as seen in the phospho-proteomic data. Figure 4, on the other hand, contains an abundance of interesting data (proteomics, phospho-proteomics, integrative analysis, target validation with small molecules, target validation with shRNA) that is all crammed into one figure. This manuscript would be more readable if the data were re-balanced between these two figures. For example, Figure 3 could explore the proteomic data, and then Figure 4 could show the follow-up validation experiments.

5. Figure 4: the integrative analysis in Figure 4F focuses on upregulated kinases. Are there downregulated kinases which fit this profile? What proteins other than kinases can be identified in this integrative manner? The focus on upregulated kinases is understandable from a therapy perspective, but the broad readership of MSB would undoubtedly be interested to know what other integrative systems-level insights can be gleaned from this data. The authors should provide additional analysis of this data.

6. Figure 7: Because MITF is not significantly expressed in cancers other than melanoma, there is no reason to test the association between MITF mRNA and CDK2 mRNA in non-melanoma cell lines and tumors. All non-melanoma cell lines from Fig. 7A and Fig. 7B can be removed altogether from the analysis (though, of course, the text in lines 363 and 364 that describe the correlation coefficient and p-value should be kept). Similarly, Fig. 7D can be removed because there is no reason to compare MITF mRNA to CDK2 mRNA in non-melanoma tumors (unless the authors can provide a justification that MITF is expressed in thyroid carcinoma (THCA)).

7. The authors should mention that Du et al, 2004 previously identified CDK2 as a drug target in melanoma. From the abstract of Du et al, 2004: "CDK2 depletion suppressed growth and cell cycle progression in melanoma, but not other cancers, corroborating previous results. Collectively, these data indicate that CDK2 activity in melanoma is largely maintained at the transcriptional level by MITF, and unlike other malignancies, it may be a suitable drug target in melanoma." In the current draft, Du et al are given credit for identifying a linkage between MITF and CDK2 expression (line 416), but not for demonstrating that CDK2 is a legitimate drug target for melanomas.

Acknowledging this previous result from Du et al does not detract from the novelty of identifying CDK2 as a drug target through integrative proteomics as found here.

Minor points

1. The title of the manuscript (as well as the running title) rightfully highlight CDK2i. However, the first mention of results in the abstract highlights "PAKs as potential targets to overcome resistance to XL888" (line 33). After this sentence, PAKs are not mentioned again in the abstract. This seems quite distracting. Why mention PAKs in the abstract if that is not the major point of your manuscript?

2. Line 97: there is a missing space between ")targets".

3. Line 98: "paradoxal" should be "paradoxical".

4. I find Figure 1A and 1B very hard to read because the graph and lines are very small. Can the authors make these figures bigger and / or find a better color scheme? The light gray cell lines (M026 and M026R) are particularly hard to read.

5. Figure 3A: the authors show a column graph of Proteins versus Phosphosites. Proteins and post-translational modifications are two different entities (because a single protein can have multiple phosphosites), and therefore, this column graph is meaningless and should be removed. Column graphs (or other graphs, for that matter) should only be employed to show the difference between two items of the same nature.

6. Line 403: should be "confidence" not "confident"

7. Supplementary Figure 3a is mentioned before (line 178) Supplementary Figures 1 and 2 (line 192). The authors should rearrange the supplementary data to avoid this.

Reviewer #2:

BRAF inhibitors are the standard of care for treating malignant melanoma. Unfortunately, drug resistance quickly develops. Therefore, clinical trials are underway to investigate combination therapies, but available biomarkers are currently lacking to help stratify patients that would benefit most from such combinations. Azimi et al. try to address a piece of this issue and discover that CDK2 may be a prime target to overcome resistance to BRAF and Hsp90 inhibitors. They take a multi-faceted approach, including thermal proteome profiling (TPP), quantitative proteomics/phosphoproteomics, in vitro assays, and analyses of publicly available datasets, to demonstrate how cell signaling is different in drug resistant cell lines and to compile evidence that point to MITF and CDK2 as potential biomarkers. Cancer biologists and clinical researchers would be interested to learn how innovative proteomic/phosphoproteomic approaches can aid in the identification of novel drug targets and potential biomarkers in the clinical trial design of combination therapy, as this paper has demonstrated. This paper has insightful experimental data, solid conclusions, and is written very well.

Major points

1. The main finding in the manuscript, targeting CDK2, was that CDK2 activity was identified in the proteomic/phosphoproteomic datasets. While TPP is an exciting technique that yielded interesting results and hypotheses, the section weakly ties into the overall CDK2 story of the paper. Can the authors elaborate as to why mentioning this technique in their story adds to the conclusions?
2. The authors' present strong in vitro proteomic and phosphoproteomic data for 2 cell lines (SK-Mel 24 and 28). One major concern was that the biology obtained, and what sets the course for the remainder of the manuscript, was defined on 1 resistant cell line. To get a better evaluation of the general mechanisms of BRAFi or HSP90i resistance, it might be useful to characterize several cell lines that display similar characteristics and then assess overlapping pathways/targets. For example, A375 DR1, MNT-1 DR100, ESTDAB037 and M026R.X1.CL all display similar responses to BRAFi and HSP90i when compared to SK-Mel 28 (Figure 1). We understand Figure 5 did address this to some degree with drug studies but none of these lines were evaluated for CDK2 or MITF protein. This would strengthen their results and support the conclusion of targeting CDK2.

Minor points

1. General cleanup of typos and grammar. (eg, In Figure 4h and legend as well as Table 2, K0386 should be K03861.)
2. In Figure 4l and 6d, the cell viability assay reported show that the viability to be as high as 180% in the Hsp90i control. Could the authors comment on the DMSO concentrations used in control versus drug? Also, why might the authors be seeing large increases in cell viability?
3. The filenames of the Supplementary Tables should be named more clearly for easier reference (ie, "Supplementary Table 1"). The files themselves contain many sheets. Therefore, creating a list or table of contents in the Supplementary Information for all of the Excel sheets inside the Excel files would also be helpful.

1st Revision - authors' response

5 December 2017

I hereby submit our revised manuscript "Targeting CDK2 overcomes melanoma resistance against BRAF and Hsp90 inhibitors" by Azimi *et al.* for publication in *Molecular Systems Biology*. We very much appreciated the reviewers' constructive comments and their recognition that the study has significant impact in the field of human melanoma.

We have carefully read the referee reports and have revised our manuscript to address these concerns. Thus, we have de-emphasized the TPP and rearranged the figures as suggested. Moreover, we have also characterized several other cell lines for the response to BRAFi and Hsp90i e.g. A375 DR1, MNT-1 DR100, ESTDAB037 and M026R.X1.CL.

Please note that Brinton Seashore-Ludlow has been listed as co-shared first author in the updated version. Furthermore, the number of figures has been decreased from 7 to 6 following the suggestion of Reviewer 1.

Below, we outline our response to each specific comment.

We believe that these clarifications and new experiments satisfactorily address the reviewers' concerns, and hope our manuscript is now suitable for publication in *Molecular Systems Biology*.

Thank you very much for your consideration.

Editorial Comments:

Dear Dr Maddalo,

Thank you again for submitting your work to Molecular Systems Biology. We have now heard back from the two referees who agreed to evaluate your manuscript. As you will see from the reports below, the referees find the topic of your study of potential interest. They raise, however, several important points, which should be convincingly addressed in a major revision of the work.

Without repeating all the issues raised by the reviewers, the major points refer to the following:

- Clarification of the way phosphosite were counted and reported

A: This has been addressed in the updated version (line 203)

- Generalization to multiple resistant cell lines (reviewer #2)

- Deemphasize the importance of the TPP profiling or clarify how it was exploited.

A: The section regarding TPP has been shortened and the results clarified in the updated version

On a more editorial level, we would kindly ask you to include a formal Data and software availability section after the Materials & Methods:

A: This has been addressed in the updated version (line 523-540)

#Data and software availability

The datasets and computer code produced in this study are available in the following databases:

- [data type]: [full name of the resource] [accession number/identifier] ([doi or URL or identifiers.org/DATABASE:ACCESSION])

- RNA-Seq data: Gene Expression Omnibus GSE46843

[<https://www.ncbi.nlm.nih.gov/geo/query/acc.cgi?acc=GSE46843>]

- Chip-Seq data: Gene Expression Omnibus GSE46748

[<https://www.ncbi.nlm.nih.gov/geo/query/acc.cgi?acc=GSE46748>]

- Protein interaction AP-MS data: PRIDE PXD000208

[<http://www.ebi.ac.uk/pride/archive/projects/PXD000208>]

- Imaging dataset: Image Data Resource doi:10.17867/10000101

[<http://doi.org/10.17867/10000101>]

- Modeling computer scripts: GitHub

[<https://github.com/SysBioChalmers/GECKO/releases/tag/v1.0>]

Reviewer #1:

MSB-17-7858

Targeting CDK2 overcomes melanoma resistance against BRAF and Hsp90 inhibitors

Azimi et al

Summary

The manuscript of Azimi et al tackles a very important question in the field of melanoma therapy: how to overcome intrinsic and acquired resistance to BRAF inhibitors (BRAFi) through judicious selection of other molecularly-targeted therapies that can be combined with BRAFi. This is particularly important given ongoing clinical trials that combine BRAFi with inhibitors of Hsp90

(Hsp90i). The authors therefore conduct a series of proteomic and phospho-proteomic experiments and integrate the data to identify CDK2 as a driver of resistance to BRAFi+Hsp90i. They then proceed to validate that targeting CDK2 with the inhibitor dinaciclib attenuates resistance to BRAFi and Hsp90i. Furthermore, the authors demonstrate the role of MITF in driving CDK2 upregulation, suggesting that MITF might serve as a biomarker for stratification of patients for CDK2 inhibitor therapy.

Overall, this is a very interesting paper dealing with a crucially important topic (intrinsic and acquired resistance to BRAFi and BRAFi combination therapies). At this time, however, there remain several points which the authors should address before the manuscript is suitable for publication in MSB.

Major points

1. The thermal proteome profiling (TPP) experiments are interesting, but they do not add significantly to the primary findings of Azimi et al regarding CDK2. These TPP experiments suggest CDK6, CDK4, PAK2, PAK4, and STAT1 as potential targets to sensitive SK-Mel-28 cells to HSPi (line 213 and Fig. 2F). Notably, CDK2 is absent from this list. In addition, the validation experiments in Fig 4C shows that STAT1i and CDK4/6i are not effective in sensitizing SK-Mel-28 to HSP90i (PAK4i did successfully sensitize SK-Mel-28 to HSP90i, but it was not nearly as effective as CDK2i or PAK1i, which had been found using alternative proteomic methods). Thus, TPP did not identify the primary focus of this article (CDK2) and was not extremely successful at predicting which targets would sensitive to HSP90i. Inclusion of the TPP data as a main figure thus distracts from the authors' primary finding that integrative proteomics identifies CDK2 as a driver of melanoma resistance to BRAFi and HSP90i (Fig. 4). This would be a much simpler and more effective manuscript if the TPP data were moved to the supplement and de-emphasized in the main text.

A: The TPP Figure with the relative data have been moved to the supplement (Figure EV1) and the TPP has been de-emphasized accordingly in the abstract and text (line 153-192).

2. In the thermal protein profiling experiments (Fig. 2f), is there a statistical value that can be used to quantify the difference between the SK-Mel-28 and SK-Mel-24 curves? Some curves look clearly different (pSTAT1 S727) but others look so very slightly different (PAK2) that the authors need to provide some statistical justification for highlighting these proteins.

A: We used the platform from Franken et al. (PMID: 26379230) that is established and used in the field. We modified the Supplemental Table (Dataset EV1) providing the p-values for the thermal proteome analysis of the proteome. For the Phosphoproteome-TPP the statistical values are provided as well and the statistical analyses were performed similarly to Franken et al, with some modifications (see Appendix Supplementary Methods).

3. Line 225: the authors report ~24,000 phosphosites of which ~15,500 have localization probability > 0.75. This threshold of 0.75 localization probability is standard for the field of phospho-proteomics and phosphosites with lower probability are typically ignored or not even reported. The problem in Azimi et al is that from reading the main text and the figure legend, it is unclear if the authors are including the phosphosites with low localization probability in their subsequent analyses. In the supplement, however, the authors have buried the information that "Only phosphopeptides with a location probability {greater than or equal to} 0.75 were considered for statistical analyses." In the interests of accuracy and simplicity, the authors should remove the mention of ~24,000 phosphosites and amend line 225 to read "we identified ~15,500 phosphosites with localization probability greater than 0.75." There is no reason to even mention the ~8,500 low confidence phosphosites.

A: This has been addressed in the updated version (line 203)

4. Figure 3 and Figure 4: Figure 3 does not add much to the manuscript because it contains only i) a technical workflow; ii) PCA to show "the quality of [the] data" (line 227); and one Western blot confirming re-activation of MAPK pathways as seen in the phospho-proteomic data. Figure 4, on the other hand, contains an abundance of interesting data (proteomics, phospho-proteomics, integrative analysis, target validation with small molecules, target validation with shRNA) that is all crammed into one figure. This manuscript would be more readable if the data were re-balanced between these two figures. For example, Figure 3 could explore the proteomic data, and then Figure 4 could show the follow-up validation experiments.

A: This has been addressed in the updated version.

The PCA has been moved to Figure EV4C

The western blot for pERK has been moved to Figure EV4D.

The proteomics and phosphoproteomics findings have been moved to Figure 2, while 'Validation of CDK2 as driver of melanoma resistance against Hsp90i and BRAFi' has been placed as Figure 3.

5. Figure 4: the integrative analysis in Figure 4F focuses on upregulated kinases. Are there downregulated kinases which fit this profile? What proteins other than kinases can be identified in this integrative manner? The focus on upregulated kinases is understandable from a therapy perspective, but the broad readership of MSB would undoubtedly be interested to know what other integrative systems-level insights can be gleaned from this data. The authors should provide additional analysis of this data.

A: We implemented Figure 2G (upper panel). We performed the identical data analysis for the downregulated entries and the unique downregulated kinases in SK-Mel 28 are ATM and CDKN2A, which are involved in cell cycle progression (line 265).

6. Figure 7: Because MITF is not significantly expressed in cancers other than melanoma, there is no reason to test the association between MITF mRNA and CDK2 mRNA in non-melanoma cell lines and tumors. All non-melanoma cell lines from Fig. 7A and Fig. 7B can be removed altogether from the analysis (though, of course, the text in lines 363 and 364 that describe the correlation coefficient and p-value should be kept). Similarly, Fig. 7D can be removed because there is no reason to compare MITF mRNA to CDK2 mRNA in non-melanoma tumors (unless the authors can provide a justification that MITF is expressed in thyroid carcinoma (THCA)).

A: Figure 7 has been modified accordingly: all non-melanoma cell lines have been removed from Figure 6A, and non-melanoma tumors have been removed from the new Figure 6B.

7. The authors should mention that Du et al, 2004 previously identified CDK2 as a drug target in melanoma. From the abstract of Du et al, 2004: "CDK2 depletion suppressed growth and cell cycle progression in melanoma, but not other cancers, corroborating previous results. Collectively, these data indicate that CDK2 activity in melanoma is largely maintained at the transcriptional level by MITF, and unlike other malignancies, it may be a suitable drug target in melanoma." In the current draft, Du et al are given credit for identifying a linkage between MITF and CDK2 expression (line 416), but not for demonstrating that CDK2 is a legitimate drug target for melanomas.

Acknowledging this previous result from Du et al does not detract from the novelty of identifying CDK2 as a drug target through integrative proteomics as found here.

A: This has been address (see line 405-406)

Minor points

1. The title of the manuscript (as well as the running title) rightfully highlight CDK2i. However, the first mention of results in the abstract highlights "PAKs as potential targets to overcome resistance to XL888" (line 33). After this sentence, PAKs are not mentioned again in the abstract. This seems quite distracting. Why mention PAKs in the abstract if that is not the major point of your manuscript?

A: PAKs have been removed in the updated version of the abstract

2. Line 97: there is a missing space between "targets".

A: This has been addressed in the updated version of the manuscript

3. Line 98: "paradoxal" should be "paradoxical".

A: This has been addressed in the updated version of the manuscript

4. I find Figure 1A and 1B very hard to read because the graph and lines are very small. Can the authors make these figures bigger and / or find a better color scheme? The light gray cell lines (M026 and M026R) are particularly hard to read.

A: This has been addressed in the updated version of the manuscript

5. Figure 3A: the authors show a column graph of Proteins versus Phosphosites. Proteins and post-translational modifications are two different entities (because a single protein can have multiple phosphosites), and therefore, this column graph is meaningless and should be removed. Column

graphs (or other graphs, for that matter) should only be employed to show the difference between two items of the same nature.

A: This has been addressed in the updated version of the manuscript and the column graph has been removed

6. Line 403: should be "confidence" not "confident"

A: This has been addressed in the updated version of the manuscript

7. Supplementary Figure 3a is mentioned before (line 178) Supplementary Figures 1 and 2 (line 192). The authors should rearrange the supplementary data to avoid this.

A: This has been addressed in the updated version of the manuscript

Reviewer #2:

BRAF inhibitors are the standard of care for treating malignant melanoma. Unfortunately, drug resistance quickly develops. Therefore, clinical trials are underway to investigate combination therapies, but available biomarkers are currently lacking to help stratify patients that would benefit most from such combinations. Azimi et al. try to address a piece of this issue and discover that CDK2 may be a prime target to overcome resistance to BRAF and Hsp90 inhibitors. They take a multi-faceted approach, including thermal proteome profiling (TPP), quantitative proteomics/phosphoproteomics, in vitro assays, and analyses of publicly available datasets, to demonstrate how cell signaling is different in drug resistant cell lines and to compile evidence that point to MITF and CDK2 as potential biomarkers. Cancer biologists and clinical researchers would be interested to learn how innovative proteomic/phosphoproteomic approaches can aid in the identification of novel drug targets and potential biomarkers in the clinical trial design of combination therapy, as this paper has demonstrated. This paper has insightful experimental data, solid conclusions, and is written very well.

Major points

1. The main finding in the manuscript, targeting CDK2, was that CDK2 activity was identified in the proteomic/phosphoproteomic datasets. While TPP is an exciting technique that yielded interesting results and hypotheses, the section weakly ties into the overall CDK2 story of the paper. Can the authors elaborate as to why mentioning this technique in their story adds to the conclusions?

A: CDK2 showed differential thermal shift in the comparison of resistant versus sensitive cells (lysate layout, Figure EV4A). This information has been implemented in the current version of the manuscript (line 178, 190). CDK2 is differentially expressed in the two cell lines, in particular it is overexpressed in the resistant cells.

Of note protein stability and protein expression levels do not necessarily correlate. This has been observed also by Savitski et al. and discusses in the First CETSA meeting held in Stockholm on 25th Sept 2017. Most likely other post-translational modifications can contribute and justify the differential stability.

2. The authors' present strong in vitro proteomic and phosphoproteomic data for 2 cell lines (SK-Mel 24 and 28). One major concern was that the biology obtained, and what sets the course for the remainder of the manuscript, was defined on 1 resistant cell line. To get a better evaluation of the general mechanisms of BRAFi or HSP90i resistance, it might be useful to characterize several cell lines that display similar characteristics and then assess overlapping pathways/targets. For example, A375 DR1, MNT-1 DR100, ESTDAB037 and M026R.X1.CL all display similar responses to BRAFi and HSP90i when compared to SK-Mel 28 (Figure 1). We understand Figure 5 did address this to some degree with drug studies but none of these lines were evaluated for CDK2 or MITF protein. This would strengthen their results and support the conclusion of targeting CDK2.

A: This has been addressed in the updated version of the manuscript (Figure EV4G). CDK2 expression levels decrease in sensitive cells upon Hsp90i treatment.

Overall, sensitive and resistant cells show opposite trends in terms of CDK2 expression levels upon Hsp90i treatment, making it a valuable target against Hsp90i tumor resistance.

Minor points

1. General cleanup of typos and grammar. (eg, In Figure 4h and legend as well as Table 2, K0386 should be K03861.)

A: This has been addressed in the current version of the manuscript (Figure 3A)

2. In Figure 4l and 6d, the cell viability assay reported show that the viability to be as high as 180% in the Hsp90i control. Could the authors comment on the DMSO concentrations used in control versus drug? Also, why might the authors be seeing large increases in cell viability?

A: DMSO concentration used in these experiments was corresponding to the Hsp90i (2:1000 dilution 0.2%). It was previously suggested that in melanoma cells, MITF and melanoma proteins are involved in sequestration of cytotoxic drugs that leads to chemoresistance (PMID: 16777967 and PMID: 23921446). On the other hand, variable MITF expression level has be correlated to different melanoma cell plasticity where low MITF level cause invasive phenotype and high MITF expression is pro-survival and leads to increased proliferation (PMID: 25866058). In this study we observed increased pigmentation and elevated MITF protein expression level upon treatment with Hsp90i in SKMEL28 cell line. Therefore, we believe that this effect is the underlying cause of increased proliferation of SKMEL28 cells upon Hsp90i.

3. The filenames of the Supplementary Tables should be named more clearly for easier reference (ie, "Supplementary Table 1"). The files themselves contain many sheets. Therefore, creating a list or table of contents in the Supplementary Information for all of the Excel sheets inside the Excel files would also be helpful.

A: This has been addressed in the current version of the manuscript

2nd Editorial Decision

9 January 2018

Thank you again for submitting your work to Molecular Systems Biology. We are now globally satisfied with the modifications made and I am pleased to inform you that we will publish your paper in Molecular Systems Biology pending the following minor amendments:

- Please include the scripts as "Computer Scripts EV1" in the form of a zip archive and a README text file at the top level and update the main text and checklist accordingly. Please note that our policies do not allow "availability upon requests" for data and scripts.
- For the HTML version of your paper, we would need the following items:
 - three to four 'bullet points' highlighting the main findings of your study
 - a short 'blurb' text summarizing in two sentences the study (max. 250 characters)
- Please reduce the number of key words to 5.
- Please replace the manuscript PDF file with a word-file.
- Please update the reference list to match the MSB reference style (listing 20 authors + et al). <http://msb.embopress.org/authorguide#referencesformat>

EV Datasets

- Please move the EV Dataset legends from the manuscript file to individual tabs in the corresponding EV Datasets.

Supplementary materials and methods

- Please move the text to the main manuscript, or rename the file Appendix and add a Table Of Content on the first page.

Callouts

- An explicit callout to Dataset EV8 missing, but there is a callout to Supplementary Table 8 on p.22. Rename it -> Dataset EV8?

Figures

- Please add a figure legend for fig 1F.

Thank you for accepting our revised manuscript "Targeting CDK2 overcomes melanoma resistance against BRAF and Hsp90 inhibitors" by Azimi *et al.* for publication in *Molecular Systems Biology*. We very much appreciated the reviewers' constructive comments and their recognition that the study has significant impact in the field of human melanoma.

We have carefully read the minor amendments and addressed them. In particular:

- We have included the scripts as 'Computer Scripts EV1'.
- We have reported three 'bullet points' highlighting the main findings of your study and a short 'blurb' text summarizing in two sentences the study at page 2 of the updated version of the manuscript provided as word-file.
- The key words have been reduced to 5.
- The reference list now matches the MSB reference style.
- The EV Dataset legends have been moved from the manuscript file to individual tabs in the corresponding EV Datasets.
- The supplementary materials and methods have been renamed as Appendix and a Table Of Content on the first page has been added.
- Supplementary Table 8 has been renamed Dataset EV8.
- A figure legend for Fig 1F has been added.

Thank you very much for your consideration.

Corresponding Author Name: Gianluca Maddalo

Manuscript Number: MSB-17-7858R